# Single cell sequencing reveals endothelial plasticity with transient mesenchymal activation after myocardial infarction

Lukas S. Tombor [1], David John[1], Simone F. Glaser[1], Guillermo Luxán [1], Elvira Forte [2], Milena Furtado[2], Nadia Rosenthal[2,3], Nina Baumgarten [1], Marcel H. Schulz[1,4,5], Janina Wittig[6], Eva-Maria Rogg[1], Yosif Manavski[1,5], Ariane Fischer[1], Marion Muhly-Reinholz[1], Kathrin Klee [5], Mario Looso [4,7], Carmen Selignow[8], Till Acker[4,8], Sofia-Iris Bibli[6], Ingrid Fleming[4,5,6], Ralph Patrick [9,10], Richard P. Harvey[9,10,11], Wesley T. Abplanalp[1,4,5,12] & Stefanie Dimmeler [1,4,5,12]✉

Endothelial cells play a critical role in the adaptation of tissues to injury. Tissue ischemia induced by infarction leads to profound changes in endothelial cell functions and can induce transition to a mesenchymal state. Here we explore the kinetics and individual cellular responses of endothelial cells after myocardial infarction by using single cell RNA sequencing. This study demonstrates a time dependent switch in endothelial cell proliferation and inflammation associated with transient changes in metabolic gene signatures. Trajectory analysis reveals that the majority of endothelial cells 3 to 7 days after myocardial infarction acquire a transient state, characterized by mesenchymal gene expression, which returns to baseline 14 days after injury. Lineage tracing, using the *Cdh5-CreERT2;mT/mG* mice followed by single cell RNA sequencing, confirms the transient mesenchymal transition and reveals additional hypoxic and inflammatory signatures of endothelial cells during early and late states after injury. These data suggest that endothelial cells undergo a transient mesenchymal activation concomitant with a metabolic adaptation within the first days after myocardial infarction but do not acquire a long-term mesenchymal fate. This mesenchymal activation may facilitate endothelial cell migration and clonal expansion to regenerate the vascular network.

[1] Institute for Cardiovascular Regeneration, Goethe University Frankfurt, Frankfurt am Main, Germany. [2] The Jackson Laboratory, Bar Harbor, ME, USA. [3] National Heart and Lung Institute, Imperial College London, London, UK. [4] Cardiopulmonary Institute, Frankfurt am Main, Germany. [5] German Center of Cardiovascular Research (DZHK), Frankfurt am Main, Germany. [6] Institute for Vascular Signaling, Goethe University Frankfurt, Frankfurt am Main, Germany. [7] Max Planck Institute for Heart and Lung Research, Bad Nauheim, Germany. [8] Department of Pathology, University Giessen, Giessen, Germany. [9] Division of Molecular Cardiology and Biophysics, Victor Chang Cardiac Research Institute, Darlinghurst, Australia. [10] St. Vincent's Clinical School, UNSW Sydney, Sydney, Australia. [11] School of Biotechnology and Biomolecular Science, UNSW Sydney, Sydney, Australia. [12] These authors contributed equally: Wesley T. Abplanalp, Stefanie Dimmeler. ✉email: dimmeler@em.uni-frankfurt.de

The endothelium forms the inner layer of blood vessels and shows remarkable plasticity and heterogeneity in structure and function, in health and disease. Endothelial cells (ECs) form a vascular plexus during early development and differentiate into arterial, venous or lymphatic fates to establish a circulatory network. At later states, endothelial cells acquire organ specific properties essential for maintaining the function of the individual organs[1,2]. Certain ECs within the endocardium undergo a process named endothelial–mesenchymal transition (EndMT), to give rise to mesenchymal cells which are necessary for proper heart and valve development[3–5]. Transitions of endothelial cells to hematopoietic cells also occurs in embryonic development[4,6]. Throughout life, endothelial cells are exposed to various mechanical, inflammatory, and metabolic environments. Phenotypic changes are facilitated by high low-density lipoprotein (LDL) cholesterol, a pro-inflammatory state and turbulent flow patterns leading to endothelial activation. This contributes to the impairment of the vasodilatory activity (so called "endothelial dysfunction"), which plays a key role in the development of atherosclerotic lesions[7]. Switches in endothelial phenotypes additionally occur during blood vessel growth, which involves a critically fine-tuned and transient specialization of endothelial cells[8]. Endothelial "tip" cells migrate at the forefront and are followed by "stalk" cells, which elongate the sprout. The tip-stalk cell phenotype dynamically switches under control of vascular endothelial growth factor (VEGF) and Notch and involves metabolic reprogramming[8–10]. Recent single-cell RNA sequencing studies provide further insights into the heterogenic gene expression signatures of ECs in tumors, leading to the identification of metabolic plasticity and collagen modification as potential critical angiogenic players[11,12]. Under pathological pro-inflammatory conditions associated with high TGF-β2 levels, endothelial cells can undergo EndMT in adulthood[4]. EndMT was reported in tumors and was suggested to contribute to atherosclerosis[13,14] and myocardial fibrosis after cardiac stress or injury[15,16]. However, recent fibroblast lineage tracings analyses suggest that the overall contribution of endothelial cells to the cardiac fibroblast population is limited[17,18]. A partial mesenchymal transition or mesenchymal activation may also contribute to (patho)physiological vessel growth by facilitating a pro-migratory and pro-invasive EC phenotype[19]. In support of a role of mesenchymal activation of ECs in vessel growth, we recently demonstrated that clonally expanding ECs in ischemic tissue express mesenchymal markers[20]. However, the molecular signatures and plasticity of endothelial cells in response to ischemia in vivo are not well established. Here we show the adaptive responses of ECs to cardiac ischemia by using single-cell technology, a powerful tool capable of deciphering individual cellular responses and transcriptional signatures within tissue.

## Results

### Kinetics of EC inflammation, survival, and proliferation.
Single-cell RNA sequencing of the non-cardiomyocyte fraction of murine hearts was performed at day 0 (homeostasis) and at day 1, 3, 5, 7, 14, 28 post myocardial infarction (MI)[21]. This revealed 19 cell clusters including 4 clusters of endothelial cells (demarcated by *Cdh5* and *Pecam*1 expression), 6 fibroblast clusters, and 7 clusters of hematopoietic cells (Fig. 1a, b and Supplementary Fig. 1a). Consistent with induction of myocardial damage, total cell numbers declined from day 1 to 7 but controls confirmed that the quality of the sequencing and cell type annotations is comparable in all samples (Supplementary Table 1, Supplementary Fig. 1b, and Supplementary Data 1). The relative abundance of endothelial cells and mesenchymal cells declined during the first week after MI, corresponding with an influx of immune cells at early time points (Fig. 1c). Significantly

augmented GO terms in EC clusters at day 1 post MI include the cellular response to hypoxia, positive regulation of inflammatory response, programmed cell death and angiogenesis (Fig. 1d). Representative genes of these pathways include the apoptosis related genes *Bax* and *Trp53*, the hypoxia induced factors *Hif1a* and *Ldha*, as well as the inflammatory cytokines *Il1b*, *Il6*, and *Tnf* (Fig. 1e). Between day 1 and 7, genes associated with epithelial mesenchymal transition, extracellular matrix organization and cellular proliferation were significantly enriched (Fig. 1d). The increase in EC proliferation was confirmed by bioinformatic analysis of cell cycle phases showing an increase in expression of cell cycle genes and number of ECs in S-phase at day 3, which is returned to homeostatic levels at day 14 after MI (Fig. 1f, g). Phospho histone H3 staining and published data[22] confirm an increased proliferation of endothelial cells at day 3 (Supplementary Fig. 2). Interestingly, the cellular response to nitric oxide was repressed between days 1 and 14 (Fig. 1d).

### ECs acquire a transient state of mesenchymal transition after MI.
Next we used single-cell trajectory analysis to identify in an unbiased manner the different states which ECs acquired after MI. We could identify 5 states that are populated by *Cdh5* and *Pecam1* expressing ECs (Fig. 2a and Supplementary Fig. 3a). MI induced an increase in ECs associated with state 3, which is a predominantly pro-inflammatory state characterized by pathways involved in neutrophil activation, inflammatory response and cytokine signaling (Fig. 2a, b and Supplementary Fig. 3a–d). The majority of ECs during the first week after MI, however, were found in state 4, which is characterized by the GO term extracellular matrix (Fig. 2b, Supplementary Fig. 3a, and Supplementary Fig. 3c). Specific genes enriched in state 4 include mesenchymal genes, cell cycle and proliferative genes, whereas transcripts involved in fatty acid up-take and signaling were down-regulated (Fig. 2b). The expression of these genes resembles the signature of EndMT or ECs undergoing mesenchymal activation. Representative upregulated genes of state 4 including *Col3a1*, *Fn1 Serpine1* and others as shown in Fig. 2c, d. Endothelial marker genes, such as *Cdh5* were down-regulated (Fig. 2c) in state 4. The induction of mesenchymal markers at day 3 after MI was confirmed by bulk RNA sequencing of isolated *Cdh5*[+] cells (Fig. 2e). Interestingly, cell numbers in state 4 were only increased between days 1 and 7 and returned to baseline levels after 14 days (Fig. 2a). A similar kinetic was observed when analyzing the total EC population (clusters 6, 7, 17, and 18) for changes in EC or mesenchymal marker expression across MI time points (Fig. 2f, g). This suggests that EC acquire a transient mesenchymal activation (EndMA) state in the first week after MI.

Using a different bioinformatic analysis strategy by re-clustering the EC population, cells were separated into ECs lacking mesenchymal markers, lymphatic ECs (cluster 9), and ECs expressing mesenchymal genes (cluster 4–6; Supplementary Fig. 4a, b). We observed a similar loss of endothelial gene expression and gain of mesenchymal genes when excluding lymphatic ECs and the other non-EC cluster 11 and 12 from the analysis (Supplementary Fig. 4c, d). At early time points (d1–d7), ECs dominantly populated in clusters with mesenchymal genes and constitute the majority of EndMA cells. Moreover, this population showed more cells in state 4 of the trajectory analysis (Supplementary Fig. 4e–g), confirming the transient nature of EndMA. Consistently, mesenchymal protein expression in endothelial cells at days 3–7 after infarction was induced as assessed by immunostaining (Supplementary Fig. 5 and Supplementary Fig. 6a, b) and FACS analysis (Supplementary Fig. 6c). Furthermore, we exclude the possibility that EndMA cells may be simply removed by cell death, since we could not find Caspase3-positive endothelial cells expressing mesenchymal marker (Sm22)

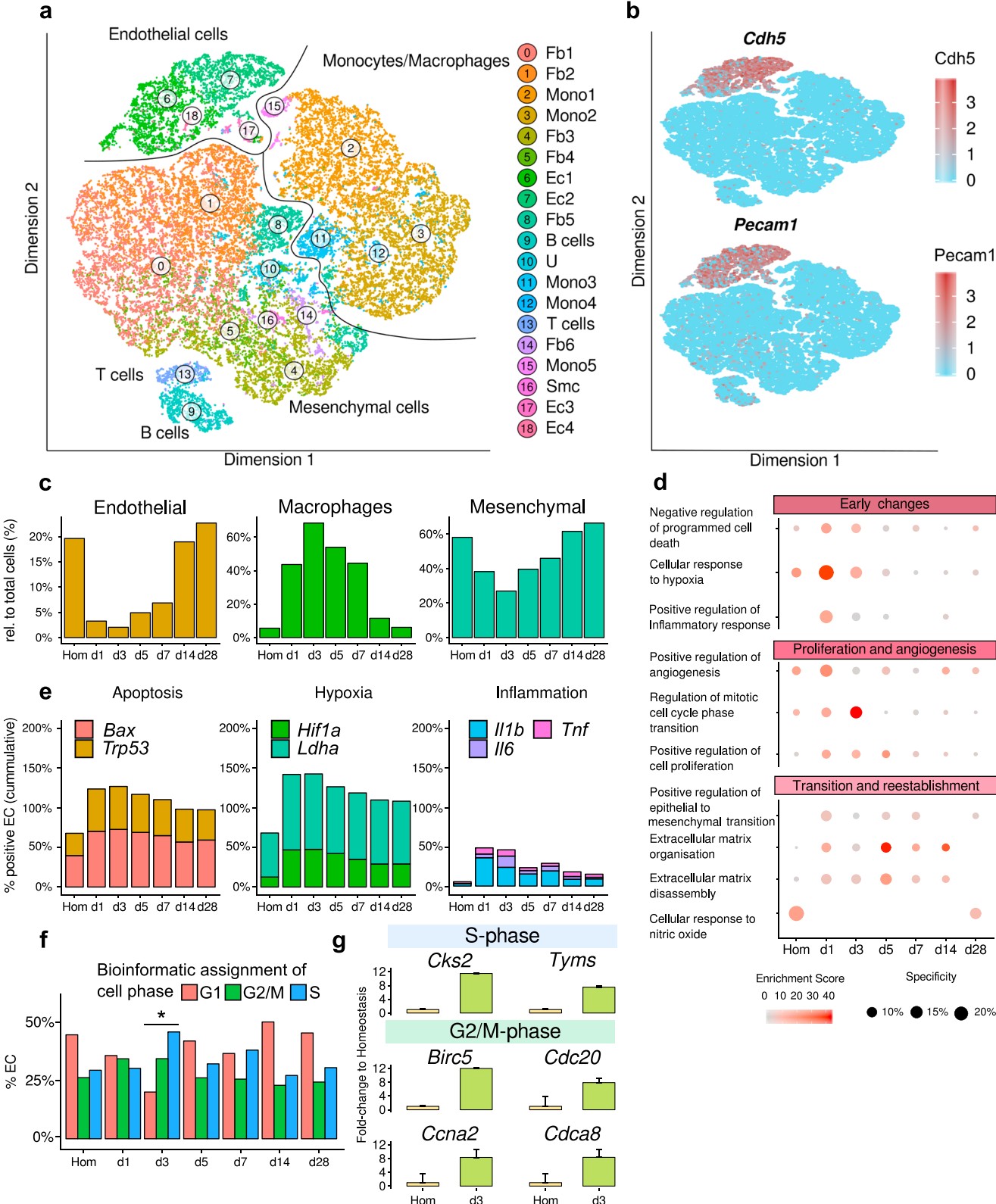

at d3 and d7 after infarction (Supplementary Fig. 7a) or enrichment for apoptosis specific gene signatures in EndMA cells (Supplementary Fig. 7b).

To provide further evidence that mesenchymal cells can revert to an endothelial cell phenotype, we performed collagen-tracing studies in *Col1a2-CreERT2;mT/mG* mice, traced at day 3 to day 7 after infarction (Supplementary Fig. 8a). Then we performed

single-cell RNA sequencing and histology at day 28. Indeed, GFP+ cells were detectable in the endothelial cluster (Supplementary Fig. 8b), demonstrating that *Col1a2* traced cells can become transcriptionally similar to endothelial cells between day 7 and day 28 after infarction. These results were confirmed by histology showing GFP expressing cells, which co-express the endothelial marker Isolectin B4 (Supplementary Fig. 8c).

**Fig. 1 Myocardial infarction induces various changes in endothelial cells. a** tSNE plot from pooled single-cell RNA sequencing data of non-cardiomyocytes from hearts after myocardial infarction (MI). Homeostasis (Hom), day(d) d1, d3, d5, d7, d14, and d28 post MI cardiovascular cells ($n = 35,312$ cells; $n = 1$ mouse per time point). We found seven cell types: fibroblasts (Fb), monocytes (Mono), endothelial cells (Ec), B-cells, T-cells, smooth muscle cells (Smc), and unknown (U). **b** tSNE plot highlighting scaled and normalized UMI counts for endothelial genes *Cdh5* (VE-cadherin) and *Pecam1* (CD31). **c** Kinetic of proportion of endothelial cells, macrophages, and mesenchymal cells relative to total number of cells per timepoint. Cell types were annotated based on Seurat's clustering in **a** and markers as described in (Supplementary Fig. 1a). Data represent $n = 1$ mouse per timepoint, $n = 4219$ endothelial cells, $n = 10,347$ macrophages, $n = 17,459$ mesenchymal cells. **d** GO-Term enrichment of genes significantly regulated (Bonferroni adjusted $p < 0.05$) over time course in endothelial cells (Pecam1$^+$/Cdh5$^+$). Color indicates combined enrichment score (based on Enrichr software). Size indicates number of significant regulated genes per total genes in GO-Term. **e** Relative number of endothelial cells expressing (UMI $\geq 1$) apoptosis marker genes (Bcl2-associated protein X; *Bax* and p53; *Trp53*), hypoxia marker genes (Hypoxia inducible factor; *Hif1a* and lactate dehydrogenase A; *Ldha*) and inflammation marker (Interleukin 1b and 6; *Il1b and Il6* and tumor necrosis factor alpha; *Tnf*). Data represent $n = 1$ mouse per timepoint, $n = 2905$ Pecam1$^+$/Cdh5$^+$ positive cells. **f** Assignment of endothelial cell phase by Seurat's 'CellCycleScoring' function. Data is presented as percentage per timepoint. *P*-value was calculated using Chi-squared test of independence with Yates correction, $p = 0.0004$. Data represent $n = 1$ mouse per timepoint, $n = 2905$ Pecam1$^+$/Cdh5$^+$ positive cells. **g** Individual fold change to homeostasis of significant (Bonferroni adjusted $p < 0.05$) markers for S-phase and G2/M-phase in d3 endothelial cells. Data shown as mean ± SEM. Data represent $n = 1$ mouse per timepoint, $n = 2905$ Pecam1$^+$/Cdh5$^+$ positive cells.

To test if mesenchymal activation may contribute to endothelial proliferation and expansion, we inhibited mesenchymal transition with the TGF-β-inhibitor Galunisertib (LY2157299)[23]. We then assessed clonal expansion of ECs after MI using Confetti mice as previously described[20]. TGF-β-inhibitor treatment reduced the incidence of clonal expansion in Confetti mice 7 days after MI in the border zone by 79 ± 19% compared to infarcted controls ($p = 0.03$; Supplementary Fig. 9), suggesting that mesenchymal activation contributes to endothelial expansion.

**Characterization of EndMA cells**. To further characterize the ECs undergoing EndMA, we compared the transcriptome of endothelial cells, which express mesenchymal markers, to endothelial cells lacking the expression of these marker genes. As expected, EndMA cells showed a high expression of mesenchymal and extracellular matrix genes (Fig. 3a), enriched expression of genes associated with epithelial mesenchymal expression (Supplementary Fig. 10a, b) and GO terms associated with extracellular matrix organization, Pdgf binding, collagen synthesis, and organization (Fig. 3b). In contrast, ECs which are not undergoing EndMA showed high expression of genes involved in fatty acid signaling, ECs enriched transcription factors and GO terms associated with VEGF and VEGF-receptor response, carbohydrate transport, Wnt signaling, vascular development, and negative regulation of EC proliferation (Fig. 3a, b). The expression of metabolism-associated genes differs between EndMA positive and negative ECs. Since EC metabolism is known to play a critical role in endothelial plasticity and EndMT[24], we performed a detailed analysis of the gene expression signatures related to glycolysis, citric acid cycle, glutamine, pentose phosphate pathway and fatty acid signaling (Fig. 3c). EndMA cells showed a high expression of glycolysis genes (Fig. 3c and Supplementary Fig. 10a), whereas the expression of fatty acid signaling, and tricarboxylic acid (TCA) cycle genes was reduced compared to ECs, which do not express mesenchymal markers (Fig. 3c), suggesting a metabolic switch from fatty acid towards a glycolytic metabolism during EndMA. Interestingly, analysis of the time course after MI demonstrates a transient switch of EC metabolism during days 1–3. Early induction of glycolysis genes at day 1 and a reduction of fatty acid signaling associated genes, such as *Fabp4* at day 3 (Fig. 3d and Supplementary Fig. 11), documented an adaptation of EC metabolism during the time frame in which mesenchymal activation occurs. In vitro studies confirmed the induction of glycolysis upon TGF-β2-stimulation of endothelial cells (Supplementary Fig. 12a). Interestingly, carbon-13 glucose tracing confirmed the increasing utilization of glucose for glycolysis without effects on glucose uptake (Supplementary Fig. 12b).

EndMA may not only change intrinsic cell functions but also may affect cell-to-cell communication. To explore whether EndMA affects the interaction of ECs with other cells in the heart, we used bioinformatic assessment of ligand-receptor relations[25]. Weighted analysis of such interactions revealed that EndMA cells showed higher ligand-interaction relations compared to EndMA negative cells (Supplementary Fig. 13a, b). EndMA cells showed increased outgoing signals involved in genes associated with the GO terms of extracellular matrix organization, Pdgf binding, and collagen fibril organization (Supplementary Fig. 13b–d). Analysis of the specific cell communication signatures between the cells revealed that EndMA cells overall have higher outgoing signals, particularly to ECs lacking mesenchymal markers (Supplementary Fig. 13a), suggesting that EndMA may affect the cell-to-cell communication in the heart after MI.

**Cdh5-lineage tracing confirms the transient nature of EndMA after MI**. Our data suggest that MI induces a transient metabolic and phenotypic switch in endothelial cells associated with EndMA. However, in the absence of lineage tracing, we may have missed ECs which fully transit to a mesenchymal phenotype and lost EC marker gene expression. To characterize the long-term fate of ECs, we used endothelial fate-mapped mT/mG reporter mice (Cdh5-CreERT2;mT/mG). Tamoxifen injection led to the expected induction of GFP in endothelial cells (Fig. 4a, b and Supplementary Fig. 14a). Analysis of GO terms associated with GFP$^+$ cells after MI showed an early increase in pathways associated with cell death, response to hypoxia, and proliferation in early stages, as well as GO terms associated with augmented extracellular matrix and epithelial mesenchymal transition at later stages (Supplementary Fig. 14b). Thus, GFP$^+$ cells showed similar gene expression signatures as the non-traced endothelial cells (Fig. 1d), supporting the hypothesis that EndMA is a transient phenomenon.

To determine the kinetics of EndMA, we assessed the expression of mesenchymal markers in GFP$^+$ cells at each timepoint. Interestingly, the number of GFP$^+$ cells expressing the endothelial marker *Cdh5* and mesenchymal markers, such as *Col1a1*, *Col3a1*, or *Serpine1*, augmented transiently with a maximum between days 3 and 7 (Fig. 4d). At later timepoints, we did not observe an increase in fully transited mesenchymal GFP$^+$ cells lacking ECs markers (Fig. 4d, right panels).

In addition, we used an unbiased approach to determine the occurrence of GFP-traced EndMA cells after MI. Re-clustering of GFP$^+$ cells led to the identification of a cluster of EndMA resembling cells (cluster 6), which showed a high expression of collagens but low expression of endothelial marker genes (Fig. 4e–g). Interestingly, the proportion of cells in cluster 6

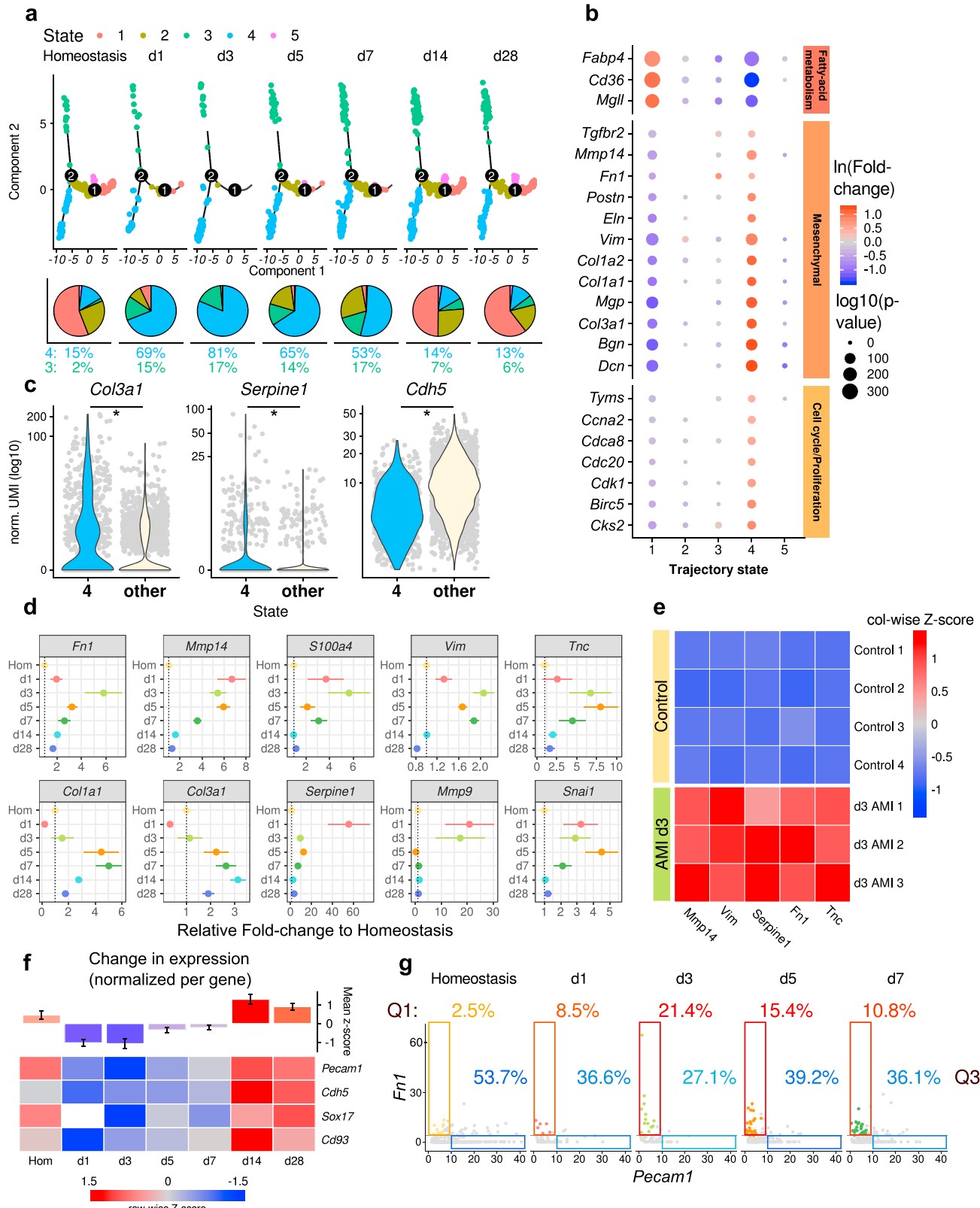

was higher at day 7 (Fig. 4g), consistent with a transient mesenchymal transition of ECs after MI.

**Mesenchymal transition is reversible in vitro.** Our data suggest that ECs transiently acquire a reversible partial mesenchymal state after MI in vivo. To gain further insights into the

reversibility of EndMT in vitro, we incubated ECs with TGF-β2 for 3 days and withdrew the stimulus. After 3 days of TGF-β2 treatment, ECs express the mesenchymal marker genes calponin (*CNN1*) and SM22 (*TAGLN*) (Fig. 4h, i, and Supplementary Fig. 15a–c). The withdrawal of TGF-β2 induced the reversion of mesenchymal marker gene expression to baseline levels within 7 days, whereas continuous incubation with TGF-β2 led to a

**Fig. 2 Endothelial cells gain mesenchymal markers at day 1–7 after myocardial infarction. a** Pseudo-time trajectory analysis of endothelial cells (ECs), using significantly regulated genes between timepoints. Pie charts showing how many cells were assigned to each cluster relative to the total number of cells per timepoint. (n = 2905 cells, ECs were selected from data shown in Fig. 1b). **b** Dot plot showing fold-change (blue to red) of marker genes associated with fatty acid oxidation, mesenchymal identity, and cell cycle proliferation between trajectory states shown in **a**. Size indicates adjusted $p$-value for each gene in $\log_{10}$ scale. **c** Violin plot showing increased values of normalized UMI counts for mesenchymal markers (*Col3a1* and *Serpine1*) and decreased values for endothelial marker (*Cdh5*) in trajectory state 4 (n = 694 cells), compared to other states (n = 2111 cells, p = $3.3 \times 10^{-76}$ (*Col3a1*), p = $2.6 \times 10^{-27}$ (*Serpine1*), p = $1.7 \times 10^{-75}$ (*Cdh5*), bimodal likelihood-ratio test, Bonferroni adjusted $p$-values) **d** EndMT marker show significant upregulation between d1–d7 in ECs compared to baseline (homeostasis). Data is presented in average fold-change to homeostasis, mean ± standard deviation. **e** Bulk-RNA sequencing analysis of isolated endothelial cells comparing homeostasis (n = 4) and d3 after acute myocardial infarction (AMI; n = 3). Color indicates column-wise z-score (blue to red). **f** Heatmap of various endothelial marker, showing reduction of mean UMI levels at d1–d7 after infarction in ECs. Upper panel represents the mean z-score of all four endothelial markers per time point. Data is presented as mean ± SEM. **g** Comparison of *Pecam1* (endothelial marker) and *Fn1* (mesenchymal marker) levels. Every point indicates an individual cell. Boxes (quadrants, Q1 and Q3) show cells which are expressing the marker strongly (top 50% of all non-zero values). Percentage values are indicative of how many cells per timepoint are grouped in the respective box. Data represent n = 1 mouse per timepoint, Homeostasis n = 592 cells, d1 n = 71 cells, d3 n = 70 cells, d5 n = 181 cells, d7 n = 277 cells.

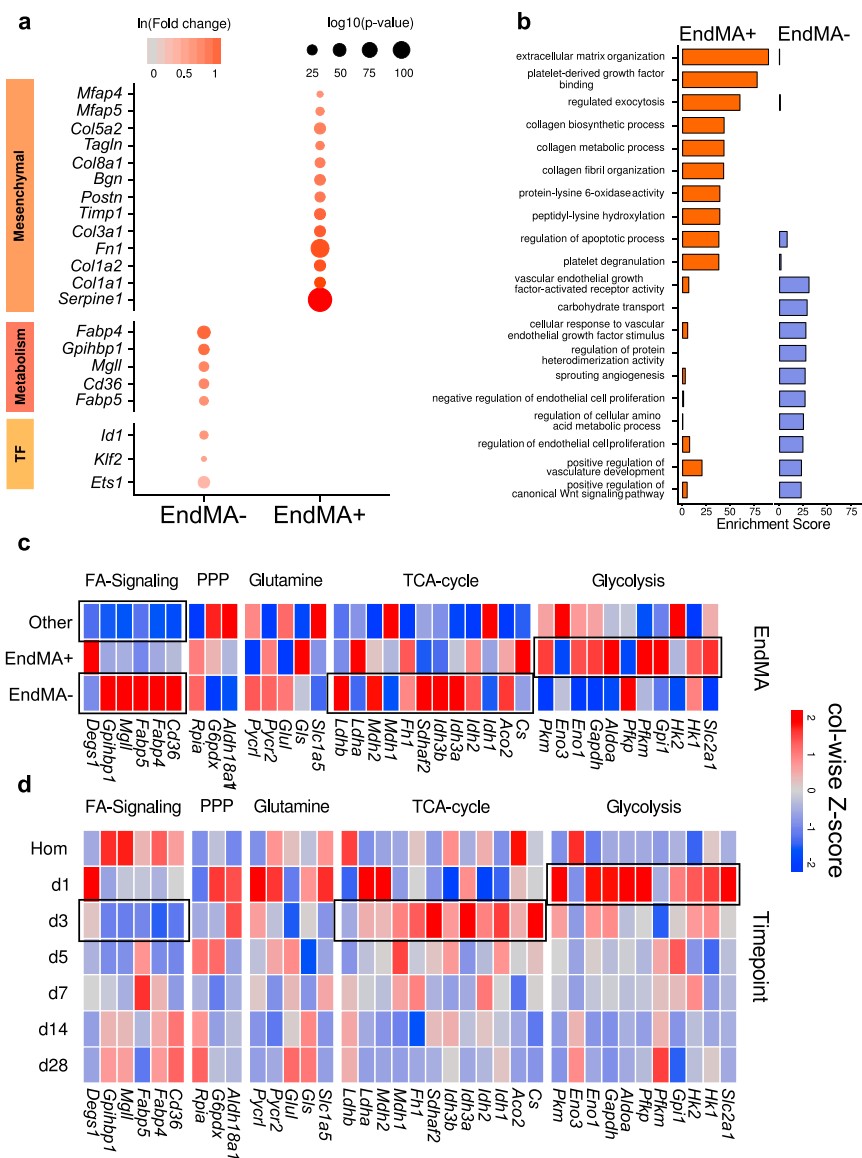

**Fig. 3 Activated endothelial cells show distinct transcriptional phenotypes. a** Dot plot showing significant change (Bonferroni adjusted $p < 0.05$) of various mesenchymal, metabolism, and transcription factor genes in *Pecam1*+/*Cdh5*+/*Serpine1*+/*Fn1*+ cells (EndMA+) compared to other endothelial cells (*Pecam1*+/*Cdh5*+/mesenchymal marker⁻; EndMA⁻) cells. Color indicates log-fold change (red to gray), size depicts significance level ($\log_{10}$ $p$-value). **b** Top 10 enriched GO-terms ranked by Enrichr's combined score for genes, significantly upregulated in EndMA+ and EndMA⁻, respectively. **c, d** Heatmap showing gene regulation of genes associated with fatty acid (FA) signaling, pentose phosphate pathways (PPP), glutamine metabolism, tricarboxylic acid (TCA) cycle and glycolysis pathway comparing EndMA+, and EndMA⁻ and all other cells in the dataset (**c**) at the indicated time points in endothelial cells (**d**). Data is presented in row-wise z-score.

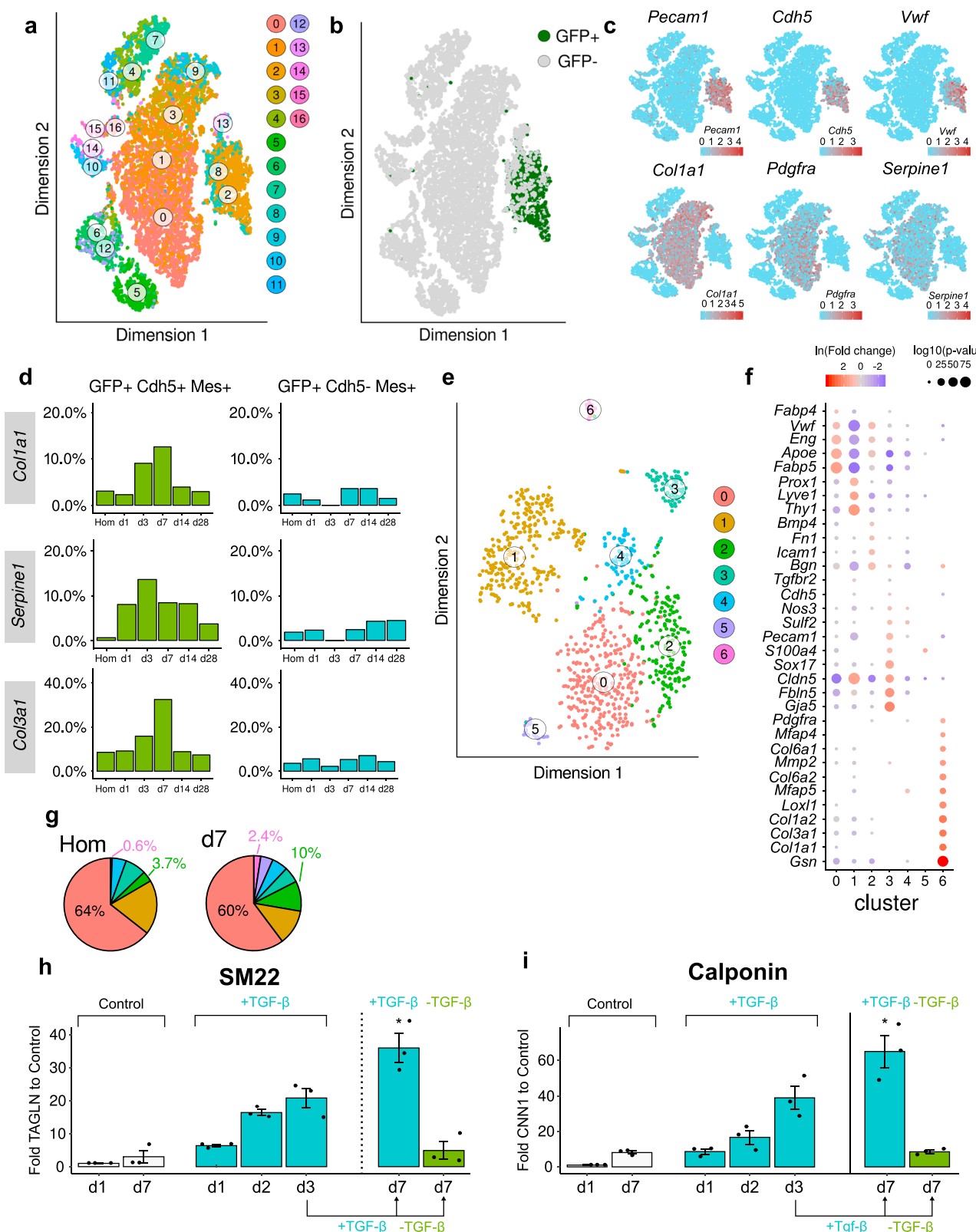

further increase in mesenchymal marker gene expression (Fig. 4h, i). Similar findings were observed when inducing mesenchymal transition by a combination of TGF-β1, IL-1b and the nitric oxide synthase inhibitors (N-Nitroarginine methyl ester; L-NAME) (Supplementary Fig. 15d, e). Notably, mesenchymal gene expression induced by prolonged incubation with TGF-β2 for up to 10 days was reversed by replacement of the stimulus to control

medium without TGF-β2 (Supplementary Fig. 15f, g). The reversible nature of EndMA was further supported by the coinciding metabolic adaptation of endothelial cells. TGF-β2 treatment induced the upregulation of glucose metabolism, which was fully reversed upon withdrawal to control medium (Supplementary Fig. 12b). To exclude that mesenchymal marker expressing endothelial cells are outcompeted by remaining endothelial cells,

**Fig. 4 EndMA is a reversible activation of endothelial cells. a** tSNE plot showing all sequenced cells for all timepoints (Homeostasis, d1, d3, d7, d14, d28) in hearts of tamoxifen treated *Cdh5-CreERT2;mT/mG* animals (*n* = 15,365 cells, 1 mouse per time point). We found four mesenchymal clusters (0,1,3,9), three endothelial clusters (2,8,13), three macrophage/monocyte clusters (4,7,11), four clusters of smooth muscle cells/pericytes (10,14,15,16), two clusters of T-cells (6,12), and one cluster of B-cells (5). **b** tSNE plot showing GFP⁺ traced endothelial cells (*n* = 1393 cells). **c** tSNE plot highlighting expression levels of endothelial marker (*Pecam1, Cdh5, Vwf*) and mesenchymal marker (*Col1a1, Pdgfra, Serpine1*) as scaled normalized UMI. **d** Left panel (green) displays relative number of cells expressing (UMI ≥ 1) GFP, *Cdh5*, and the mesenchymal markers (*Col1a1, Serpine1*, or *Col3a1*) per timepoint. Right panel (blue) shows population of cells not expressing *Cdh5* in this context. Data represents *n* = 1 mouse per timepoint. Homeostasis *n* = 592 cells, d1 *n* = 87 cells, d3 *n* = 44 cells, d7 *n* = 166 cells, d14 *n* = 304 cells, d28 *n* = 270 cells. **e** tSNE plot of reclustered GFP positive cells, showing 7 independent clusters. **f** Gene expression of different marker genes in different cluster shown in **e**. Color indicates the log-fold change (blue to red), size shows log *p*-value. **g** Relative number of cells assigned to the different clusters from (**e**) in homeostasis and d7. **h, i** Fold change of SM22 (*TAGLN*) (**h**) and Calponin (*CNN1*) (**i**) mRNA levels measured by qRTPCR (*n* = 3 independent experiments) in HUVECs treated with TGF-β2 supplemented medium (turquoise) and control medium (gray). Expression values were normalized to *RPLP0*. HUVECs cultured for 3 days in TGF-β2 and subsequent cultivation in control medium until 7 days are shown in green. *P*-value was calculated using Kruskal–Wallis test (*p* = 0.005 SM22, *p* = 0.006 CNN1). Data shown as mean ± SEM.

we additionally treated TGF-β2 stimulated cells with a cell cycle inhibitor and assessed the effect on the reversibility. However, cell cycle inhibition did not affect reversibility of EndMT (Supplementary Fig. 16).

To assess if TGF-β2 treatment leads to a true fate change, we analyzed DNA methylation of mesenchymal genes. We did not observe changes in DNA methylation levels in gene bodies within the mesenchymal genes *COL4A1*, *TAGLN*, and *CNN1* after TGF-β2 stimulation and withdrawal (Supplementary Fig. 17a–c). However, we observed a reversible demethylation of regulatory regions upon TGF-β2 stimulation (Supplementary Fig. 17d). These observations confirm that mesenchymal gene expression induced by several different protocols is reversible and is associated with changes in DNA methylation of regulatory regions.

## Discussion

The presented study provides a detailed analysis of EC gene expression signatures after cardiac ischemia. We demonstrate that ECs undergo a transient mesenchymal activation that is associated with profound metabolic adaptations (Supplementary Fig. 18). The transient nature of mesenchymal gene expression in endothelial cells was supported by Cdh5-lineage tracing studies, which demonstrated that Cdh5-traced cells transiently express transcripts typically associated with EndMT. Mesenchymal transcript expression in Cdh5-traced cells returns to levels detected in homeostasis condition after more than 10 days post-myocardial infarction. Interestingly, the majority of mesenchymal gene expressing GFP⁺ traced cells retain a low expression of endothelial genes (Fig. 4d), suggesting that they undergo a transient EndMA rather than full EndMT. In support of a partial transition, our data show that most EndMA cells express modest levels of mesenchymal marker genes in comparison to *bona fide* fibroblasts (Supplementary Fig. 19). Consistent with our data demonstrating that only few GFP-traced cells fully lose their endothelial signature and express high levels of mesenchymal genes, recent single-cell RNA sequencing studies also failed to detect significant numbers of cells undergoing a full mesenchymal transition after myocardial infarction[26]. Initial studies have imposed the concept, that the conversion of endothelial cells to mesenchymal cells was a permanent shift between two differentiated cellular states. However, in the context of cardiac ischemia our study suggests that EndMA likely represents a reversible continuum in response to a hypoxic and inflammatory injury environment, instead of a differentiation process in its classical sense. This model is consistent with our in vitro study clearly documenting a reversible nature of EndMA after removal of EndMT promoting stimuli. Interestingly, similar findings on endothelial cells were already reported in 1992, demonstrating that withdrawal of TGF-β1 after 10 days incubation resulted in the re-appearance of polygonal cells with an endothelial phenotype[27]. However, the mechanisms by which this switch back to an endothelial phenotype

occurs and if this includes epigenetic changes is unclear. Our in vitro experiments utilizing endothelial cells did not show changes in DNA methylation patterns at promoters of mesenchymal genes after TGF-β2 stimulation, suggesting that a true transition to a mesenchymal phenotype driven by epigenetic changes at the level of DNA methylation is a rare event under these conditions. We also did not observe a hypermethylation of *RASAL1*, which was shown to be induced after long-term treatment of ECs with TGF-β1 for 12 days[28]. Instead, we found a reversible methylation pattern at several regulatory elements, but the potential functional implications of those methylation events need to be studied in more detail.

What is the physiological or pathophysiological relevance of this transient EndMA? The partial EndMA may contribute to new blood vessel growth, by promoting a pro-migratory and pro-invasive state. Indeed, EndMA cells show signatures of genes associated with extracellular matrix organization, migration and proliferation, and the inhibition of TGF-β signaling reduced expansion of endothelial cells. Reports of the literature also support such a model, as it has been demonstrated that inhibition of EndMT driving transcription factors such as *Snail* or *Slug* prevents EC migration and in vitro angiogenesis[29] and clonally expanded ECs showed a high expression of EndMT marker genes[20]. On the other hand, even the partial induction of EndMA may have detrimental effects. Although, our studies suggest that the direct contribution of EndMA to cardiac fibrosis by transition of ECs to cardiac fibroblasts is limited, we cannot exclude that cells undergoing EndMA may secrete extracellular matrix and inflammatory proteins, which affect the wound healing response and may thereby indirectly contribute to fibrosis. A transient mesenchymal activation may also contribute to other pathological states in vascular disease, such as atherosclerosis. As mesenchymal transition is associated with cellular detachment[30] one might envision a role in the acute events leading to plaque erosion.

Together, our study demonstrates a transient mesenchymal activation of endothelial cells and may clarify some controversies in the literature. While our RNA single-cell sequencing study clearly confirms the existence of mesenchymal marker expressing cells, and therefore supports the many studies reporting endothelial-mesenchymal transitions[31], this process appears reversible and therefore was not found to occur in long-term fate studies using lineage tracing. Future investigations may provide insights as to what extent EndMA (as opposed to EndMT) occurs in other injury or disease conditions and how risk factors affecting the stability of the endothelial epigenome may be able to interfere with the reversibility of the process.

## Methods

**Animals.** All animal experiments were executed in agreement with the animal welfare guidelines and German national laws. All animal experiments and study protocols were authorized by the competent authority (Regierungspräsidium Darmstadt, Hessen, Germany). Mice were held at 23 °C ambient temperature and

60% humidity in 10 h/14 h light/dark cycle. C57BL/6 J *Cdh5-CreERT2* mice (obtained from The Jackson Laboratory, MGI:3848982) or C57BL/6 J *Col1a2-CreERT2* mice (obtained from The Jackson Laboratory, Stock No. 029235), were bred with *mTomato/mGFP^fl/fl* (mT/mG) mice (obtained from The Jackson Laboratory) to generate *Cdh5-ERT2^+/-;mT/mG* or *Col1a2-CreERT2^+/-;mT/mG* mice. For *Cdh5-ERT2^+/-;mT/mG*, mice were used at an age of 10–12 weeks (*n* = 6, male mice). Cre expression was induced by daily intraperitoneal injections of 2 mg/mouse tamoxifen (T5648; Sigma Aldrich) for 1 week in all mice. One week after tamoxifen injection, left anterior descending (LAD) coronary artery ligation surgery was performed, inducing myocardial infarction (MI) as previously described[32]. The animals were anesthetized with isoflurane and analgesia was delivered by an intraperitoneal injection of Buprenorphin (0.1 mg/kg body weight (BW)) in combination with a block of the intercostal nerves by using Bupivacaine (1 mg/kg BW, 0.25% Bupivacaine). For postoperative analgesia, Buprenorphin and Caprofen (5 mg/kg BW) were given every 12 or 24 h for 3 days. Postoperative infection was prevented by giving Ampicillin (100 mg/kg BW) via drinking water. Under mechanical ventilation, the MI was induced by permanent ligation of the left anterior descending coronary artery. Echocardiography was performed on day 0 to ensure the induction of a MI using the program VevoLab (Visualsonics). After days 1, 3, 7, 14, or 28 post MI, hearts were harvested for single-cell RNA sequencing. For *Col1a2-ERT2^+/-;mT/mG*, LAD surgery was performed and tamoxifen was injected from 3–7 days post infarction. Hearts were collected 28 days after infarction. For immunofluorescence stainings, we used wildtype BL6 C57BL/6 J animals (without tamoxifen treatment) undergoing the same experimental procedure.

For Galunisertib LY2157299 (A11017, AdooQ) in vivo treatment, we injected tamoxifen in *Confetti^fl/wt-Cdh5-CreERT2* and induced MI 6 days after last treatment[20]. Formulations were prepared in vehicle buffer containing 0.5% carboxy-methyl cellulose (3333.1, Carl Roth). Suspensions were administered per oral gavage (75 mg/kg) 2 h before, 24 h after MI and then twice a day. Seven days after MI, hearts were harvested, as described above.

**Single-cell RNA sequencing.** For the first analysis shown in Figs. 1–3, we used a publicly available data set[21]. For the lineage tracing studies, hearts were digested with Multi Tissue Dissociation Kit 2 (130-110-203, Miltenyi), as described for application of dissociation of adult mouse heart. In brief, after flushing murine hearts with HBSS, hearts were harvested, immediately minced and transferred to an enzyme mix and dissociated using the gentleMACS Dissociator as described in the manufacturer's protocol. Samples were filtered through a MACS SmartStrainer (70 μm) with 3 mL of PBS with 20% FBS. Samples centrifuged and the resultant supernatant aspirated. The cell pellet was resuspended in PEB buffer and 1× Red Blood Cell Lysis Solution to remove erythrocytes. Following erythrocyte removal, Enzyme A of the Multi Tissue Dissociation Kit 2 was added to PBS in a fresh tube and then combined with the cell pellet. Following this step, cells were resuspended in Dead Cell Removal MicroBeads (130-090-101, Miltenyi). Samples were mixed and incubated following Miltenyi's protocol instructions. We made the positive selection with a column of type MS (for up to $10^7$ dead cells and up to $2 \times 10^8$ total cells). Prior to its use, the column was rinsed with binding buffer. Effluent was collected as the live cell fraction. Cells were counted using Trypan blue and a Neubauer chamber. Cells were then resuspended appropriately for loading on the 10X Genomics Chromium Controller chip. Resulting non-cardiomyocyte cellular suspensions were loaded on a 10X Chromium Controller (10X Genomics) according to manufacturer's protocol. Murine scRNA-seq libraries were prepared using Chromium Single Cell 3′ v3 Reagent Kit (10X Genomics), according to manufacturer's protocols. Briefly, the initial step consisted in generating cell partitioning droplets where individual cells were isolated together with gel beads coated with unique primers bearing 10X cell barcodes, UMI (unique molecular identifiers) and poly(dT) sequences. Reverse transcription reactions were engaged to generate barcoded full-length cDNA followed by the disruption of emulsions using the recovery agent and cDNA clean up with DynaBeads MyOne Silane Beads (37002D, Thermo Fisher Scientific). Bulk cDNA was amplified using a Biometra Thermocycler Professional Basic Gradient with 96-Well Sample Block (98 °C for 3 min; cycled 14×: 98 °C for 15 s, 67 °C for 20 s, and 72 °C for 1 min; 72 °C for 1 min; held at 4 °C). Amplified cDNA product was cleaned with the SPRIselect Reagent Kit (B23318, Beckman Coulter). Indexed sequencing libraries were constructed using the reagents from the Chromium Single Cell 3′ v3 Reagent Kit as follows: fragmentation, end repair and A-tailing; size selection with SPRIselect; adapter ligation; post-ligation cleanup with SPRIselect; sample index PCR and cleanup with SPRI select beads. Library quantification and quality assessment was performed using Bioanalyzer Agilent 2100 using a High Sensitivity DNA chip (5067-4627 Agilent Genomics). Indexed libraries were equimolarly pooled and sequenced by GenomeScan (Leiden, Netherlands).

Single-cell RNA-seq outputs were processed using the CellRanger (10X Genomics) suite versions 3.0.1. For mapping reads from tracing experiments we modified the mm10 (GRCm38.p4, version 1.2.0) reference genome and added sequences of *EGFP* and *tdTomato*, obtained from the original vector which had been used to create the mouse line[33], as part of a virtual non-existing chromosome. We annotated these sequences in the GTF (gene transfer format) file and rebuild the reference genome by the *mkref* command implemented in the CellRanger software. We confirmed the specificity by mapping to data obtained from wildtype animals or either tamoxifen injected or non-injected *Cdh5-CreERT2*.

Raw base count files were demultiplexed and processed by *mkfastq* implemented in the CellRanger software. Raw reads were then mapped to the customized reference genome by using CellRanger *count*. Secondary analysis was conducted using the Seurat 2.3.4 package in R[34]. We filtered the data for cells expressing at least 200 genes, as suggested by the distributer's tutorial (satijalab. org). In order to avoid analyzing doublets or dead cells, we removed barcodes with a high mitochondrial content (>10% of total UMI) as well as barcodes with abnormal high (upper 5% of all cells) or low (lower 1% of all cells) number of UMI or genes, respectively. The remaining matrix was log-normalized and scaled.

Genes that were detected as variable in each sample, were then used for alignment of subspaces in a canonical correction analysis (CCA) as implemented in Seurat. Number of subspaces were estimated by the "MetageneBicorPlot" function. We aligned 25 subspaces and set the resolution parameter of *FindCluster* to 0.6. We used t-distributed stochastic neighbor embedding (t-SNE) to visualize cell clusters.

Cell cycle states of each cell were predicted by the algorithm implemented in Seurat[35]. The cell cycle score was based on mouse orthologous genes to the default human genes.

For building endothelial subsets based on expression of marker genes we filtered cells which express *Cdh5* and *Pecam1* (UMI > 0). In data from lineage tracing experiments we used a dynamic threshold to separate EGFP positive cells. Based on the density distribution of EGFP normalized UMIs for each sample we calculated a cutoff that separated real GFP expressing cells (population with highest EGFP normalized UMI counts) from false positive based on a local minimum. We successfully validated this method and estimated the number of GFP positive cells for a sample which had not been injected with tamoxifen at almost 0.

Gene ontology enrichment analysis have been performed with Enrichr[36] (R package version 1.0.) We used *GO_Biological_Process_2018* and *GO_Molecular_Function_2018* distributed via the Enrichr webpage and provided from the gene ontology consortium as our reference gene-set libraries. GO terms have been ranked and displayed by their "Combined Score" as implemented in Enrichr, which combines the z-score of the term's deviation to it's expected rank and the p-value from Fisher's exact test and their specificity score as the fraction of genes enriched to the total number of genes in the respective GO term. Gene set enrichment analysis was performed using clusterProfiler (R package version 3.10.1) and gene sets from Molecular Signatures Database (MSigDB, gsea-msigdb.org)

For visualizing possible sub-states of endothelial cells, we applied trajectory analysis to our data, identifying pseudotemporal ordering of cells along the actual time course. We therefore used the Monocle (Version 2.6.4) toolbox and followed the standard pipeline, as described in their tutorial[37]. In brief, Seurat objects were converted into Monocle using the "importCDS" function and size factors and dispersion were estimated. We then defined genes as expressed, when exceeding a threshold of 0.1 and where at least expressed in 10 cells. We then based our trajectory on differential genes between the individual timepoints, as calculated with "differentialGeneTest". Cells where then accordingly filtered using "setOrderingFilter". Pseudotime reduction of dimension was applied with the "DDRTree" method, as implemented in the "reduceDimension" function. Finally, we ordered cells by using "orderCells" and plotted trajectories split by state and timepoint.

Ligand-receptor interaction analysis was performed on transcriptomic profiles[25]. Briefly, a weighted directed graph with four layers of nodes was built linking source cell types (layer 1), defined by expression of a ligand (layer 2), to target cell types (layer 4) expressing a corresponding receptor (layer 3), after reference to a curated map of ligand-receptor pairs. Source-ligand and receptor target edges were weighted according to expression fold-change in ligands and receptors, respectively. Ligand-receptor edges were further weighted by mouse-specific association scores from the STRING database. Permutation testing (100,000 permutations) of randomized network connections was applied to determine significant source-target network connections following Benjamini–Hochberg multiple-testing correction (adjusted *p* < 0.01).

**Immunohistochemistry.** Male C57BL/6 J mice were sacrificed by cervical dislocation on d3 (*n* = 6) and d14 (*n* = 5) post LAD surgery. Age-matched non-operated animals (homeostasis) have been sacrificed on the same day (*n* = 4, respectively). Hearts were flushed with HBSS (14025-050, Gibco) by injection into the *Vena cava caudalis*, cut in half transversally and washed in HBSS and DPBS (14190-094, Gibco). Tissue was fixed using 4% Paraformaldehyde (1.04005-1000, Merck) 4 °C on an orbital shaker overnight. Hearts were cryopreserved using increasing dilutions of 10%, 20% and 30% sucrose (S0389-1kg, Sigma) and 1% Polyvinylpyrrolidon (PVP, P-5288, Sigma) in PBS and incubated for 24 h at 4 °C on orbital shaker, respectively. Hearts were embedded in embedding moulds (58951, Thermo Scientific), using 15% sucrose, 1% PVP and 8% gelatine (G1890-100G, Sigma). Tissue was frozen at −80 °C overnight and cut in 50 μm sections. Sections were thawed at room temperature for 30 min. Antigen retrieval was performed by boiling the sections for 15 min in citrate buffer (pH 6, 0.1 M). Quenching of endogenous peroxidases was performed in 100% Methanol with 1% $H_2O_2$ for 40 min. Sections were blocked with 3% BSA (0163.2, Roth), 5% donkey serum (7475, abcam), 20 mM $MgCl_2$, and 0.3% Tween 20 (P1379, Sigma). Primary antibodies were incubated at 4 °C overnight followed by secondary antibody incubation for 1 h at room temperature.

For GFP stainings, we used hearts isolated from *Cdh5-CrERT2;mT/mG* or *Col1a2-CrERT2;mT/mG* reporter mice and embedded them in O.C.T medium

(4583, Tissue-Tek®). Hearts were cut transversally in 10 μm sections and frozen at −80 °C and stained as described above with respective antibodies.

For HUVEC stainings, cells were cultured in μ-slide eight-well chambers (80826, Ibidi) coated with 1 μg/mL human fibronectin (F0895, Sigma Aldrich). After treatment, cells were pre-fixed for 2 min by adding equal amounts of 4% PFA to the cultivation medium and subsequently fixed in 2% PFA for 20 min at room temperature. We permeabilized and blocked cells for 1 h in blocking buffer containing 5% donkey serum, 0.3% Triton X-100, 2% BSA. Primary antibodies were incubated overnight in blocking buffer at 4 °C. We incubated secondary antibodies for 1 h on room temperature in dilution buffer, containing 1% BSA, 1% donkey serum, and 0.3% Triton X-100.

For *Confetti^{wt/fl}-Cdh5-CreERT2* animals, hearts were fixed by injection of 4% PFA into the *Vena cava caudalis* and subsequent fixation in 4% PFA overnight at 4 °C. Tissue was embedded in O.C.T medium, cut transversally in 10 μm sections and analyzed without further staining.

Used antibodies are described in the supplement (Supplementary Table 2). Images were taken under a Zeiss LSM780 or Leica SP8 confocal microscope.

**Cell culture**. The in vitro experiments were performed using human umbilical vein endothelial cells (HUVECs) purchased from Lonza. HUVECs were cultured in endothelial basal medium (CC3121, Lonza) with all recommended supplements (hydrocortisone (CC-4035C, Lonza), ascorbic acid (CC-4116C, Lonza), bovine brain extract (CC-4092C, Lonza), epidermal growth factor (CC-4017, Lonza), 10% fetal bovine serum (FBS) (CC-4101, Lonza) and Gentamicin + Amphotericin-B-1000 (CC-4081C, Lonza)). Cells were incubated at 37 °C and 5% $CO_2$.

**Endothelial-to-mesenchymal transition assay**. For the induction of endothelial-to-mesenchymal transition (EndMT), HUVECs were cultured either in the normal cell culture medium as described in the section "Cell Culture" (Control), or in a medium without epidermal growth factor and bovine brain extract, supplemented with either 10 ng/mL TGF-β2 (302-B2, R&D System), or 10 ng/mL TGF-β1 (240-B, R&D System) and 10 ng/mL IL-1β (200-01B, Peprotech)[38]. Nitric oxide synthase was inhibited by 1 mM N-Nitroarginine methyl ester (L-NAME). To reverse EndMT induction, the medium containing the named stimuli or inhibitor was replaced after 3, 7, or 10 days of culture with normal control medium and cultivated for 4 days. Medium and treatment was replenished every second day. After day 1, 2, 3, 7, 10, 11, or 14 RNA was isolated. For cell cycle inhibition assay, we treated the cells with 2.5 μM Vitexicarpin (CFN98172, Chem Faces) or equal amounts of DMSO after TGF-β2 treatment for 4d.

**RNA analysis**. RNA was isolated using the miRNeasy-kit (217004; Qiagen). Additional DNase I (79254; Qiagen) digestion according to the manufacturer's protocol was performed. 1 μg of RNA from each sample was reverse-transcribed by MMLV Reverse Transcriptase (N8080018; Life technologies), using random hexamer primers (10 min at 25 °C, 15 min at 42 °C, 5 min at 99 °C) as previously described[39]. We performed qPCR analysis using Fast SYBR Green Mastermix (4385612, Life Technologies) and the ViiA7- Realtime qPCR System (Life Technologies). Expression levels were normalized to the housekeeping genes *RPLP0* and *GAPDH* and analyzed using the $2^{-\Delta Ct}$ method. The sequences of the used primers are listed in the supplement (Supplementary Table 3).

**FACS Analysis**. For FACS analysis, we used hearts from *Cdh5 CreERT2;mT/mG* and processed them as described above for single-cell sequencing. Cells were then fixed for 20 min in methanol on ice and blocked in 3% BSA, 5% donkey serum and 0.1% Triton-X100. We incubated cells with primary antibodies (Supplementary Table 2) for 20 min on ice and with secondary antibodies for 30 min on ice. Additionally, FACS was performed on 1 h 1 mM BrdU treated HUVECs, using a BrdU kit (559619,BD Pharmingen). Here, cells were fixed and permeabilized and treated with DNAse to expose BrdU labeled DNA. Cells were stained using anti-BrdU antibody and 7-AAD dye. All reagents supplied by the kit. Analysis was performed on BD LSRFortessa X-20 using FACS Diva software.

**Active glycolytic flux**. HUVECs were seeded on a fibronectin coated Seahorse XF96 culture plate (Agilent Technologies) at a density of $6 \times 10^4$ cells per well in the respective endothelial growth medium. The medium was changed to Krebs Henseleit buffer (111 mM NaCl, 4.7 mM KCl, 1.25 mM $CaCl_2$, 2 mM $MgSO_4$, 1.2 mM $Na_2HPO_4$) containing 1 mM sodium pyruvate 1 h prior to the measurement with a XFe 96 extracellular flux analyzer (Agilent, Technologies). Activators and inhibitors were used at the following final concentrations: 10 mM glucose, 1.5 μM oligomycin A, 100 mM 2-deoxy-D-glucose. Each measurement was averaged from three technical replicates of three different biological replicates.

**^{13}C Glucose metabolic flux analysis**. HUVEC were seeded in 10 cm culture dishes and treated with the respective media. In all, 48 h prior to collection, media were substituted with glucose free media, supplemented with 13 C uniformly labeled glucose (Sigma) in a final concentration of 17 mM. Samples were collected and prepared as previously described[40]. LC-MS/MS experiments were performed on an Agilent 1290 Infinity UPLC system (Agilent, Waldbronn, Germany) coupled

to a QTrap5500 mass spectrometer (Sciex, Darmstadt, Germany), equipped with an ESI TurboIonSpray source. The flow rate was 0.22 mL/min, autosampler temperature was set at 6 °C, and the column compartment was set to 35 °C. Separations were performed on an Asahipak NH2P-40 (250 × 2 mm, 4 μm particle size; Showa Denko, Munich, Germany). The mobile phase was composed of 20 mM ammonium carbonate in H2O/5% acetonitrile (ACN), pH = 10 and ACN. After an initial 3.5 min isocratic elution of 99.9% ACN, the percentage of ACN decreased to 85% at 3.6 min, to 75% at 8.1 min, to 0% at 14 min, back to 99.9% at 34 min. The composition was maintained at 99.9% ACN until the 42 min. The QTrap5500-MS system was operated in triple quadrupole mode wit positive/negative ion switching. Ion spray voltage of 5500/-4500 V were applied, respectively. Curtain gas was set to 30 psi, collision gas to medium, source temperature to 500 °C, ion source gas 1 to 35 psi, ion source gas 2 to 35 psi. Analyst 1.6.2 and MultiQuant 3.0 (both from Sciex, Darmstadt, Germany), were used for data acquisition and analysis, respectively.

**DNA methylation**. For assessment of DNA methylation, HUVECs were cultured as described above and DNA was isolated using the QIAamp DNA Micro Kit (56304, Qiagen), according to the manufactures protocol.

DNA concentration was quantified using the Qubit dsDNA HS Assay kit (Q32851, Invitrogen). 200–500 ng of DNA was used as input for DNA methylation analysis. The Infinium Human-Methylation EPIC BeadChip (850k) (WG-317, Illumina) was used to determine the DNA methylation status of more than 850.000 CpG sites, respectively following the producer's guidelines. On-chip quality metrics of all samples were carefully controlled.

Infinium human methylation EPIC array analysis was performed by using RnBeads (vesion 2.2)[41] and ADMIRE[42]. Raw intensities were taken from IDAT files and quality control was performed on all probes and samples. During preprocessing, probes and samples have been filtered for SNP-enrichment and highest impurity using Greedycut (FDR adjusted $p < 0.05$). The data was normalized using *dasen* method[43]. Individual site-based methylation levels were calculated and subsequently used for region-based methylation level assessment (CpG island/promoter/ tails/gene body). Differential methylated regions were determined by comb-p method[44] and the combined rank score of RnBeads. For illustration of beta values ADMIRE visualization functions were used.

For regulatory element analysis, 848,254 CpG probes that were retained after EPIC array quality filtering were overlapped with 2.4 million annotated regulatory elements (REMs) of the EpiRegio database[45] and analyzed using RnBeads software. CpG methylation was averaged over all CpG probes overlapping a REM. Differentially methylated regulatory elements with a mean methylation difference ≥0.1 between Control and +TGF-β2 (d3) or +TGF-β2 (d3) -TGF-β2 (d7) respectively, and FDR corrected p-values ≤ 0.05 were selected. These significantly differentially methylated REMs were clustered by their DNA methylation pattern using hierarchical clustering.

**Statistics and reproducibility**. For single-cell RNA Sequencing statistical analysis of differential gene expression was performed with Seurat's FindMarkers or FindAllMarkers functions by using a bimodal likelihood estimator test suitable for zero-inflated scRNA-seq data[46]. Data is always shown in p-values corrected for multiple testing (Bonferroni).

For GO-Term enrichment of significantly regulated genes we used enrichR software suite[36]. We used the combined score to report on enrichment, which combines the term's deviation to its expected rank and the p-value of Fisher's exact test.

For qPCR data, ANOVA non-parametric Kruskall–Wallis multiple comparison test with posthoc Dunn´s test was used for comparison of multiple groups or a two-sided Student's $t$ test with Welch's correction. A value of $p < 0.05$ was considered statistically significant.

We used a $\chi^2$-test of independence to indicate significant changes in ratios of cell cycle assignment and considered $p < 0.01$ as significant.

Representative images shown have been chosen from $n = 3$ independent experiments.

For *Confetti;Cdh5-CreERT2* analysis we separated each border zone image ($n = 5$ per animal) into 6 same size areas and counted the occurrence of colored cells. Areas with clonal expanding cells, which had more than two times the standard deviation of the animal mean of counted RFP, YFP and GFP cells, were compared between the groups. We used Fisher's exact test for statistical analysis of the data.

**Reporting summary**. Further information on research design is available in the Nature Research Reporting Summary linked to this article.

## Data availability

The single-cell RNA−seq datasets generated in this study are available at ArrayExpress (E-MTAB9816 and E-MTAB9817). Bulk-RNA sequencing has been deposited at Sequencing Read Archive (PRJNA679225). RnBeads output files are available (10.6084/m9.figshare.13247210.v1). All other data are included within the article, source data, supplementary data or can be made available upon request. Source data are provided with this paper.

## Code availability

Codes for R analysis of data presented in this study are available on GitHub (github.com/TLukas1/NatComm_Tombor_et_al_2020).

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

## Acknowledgements

We wish to thank Felix Vetter, Maximilian Merten, Lisa-Maria Kettenhausen and Katrin Häfner for offering technical assistance to the authors of this study. We acknowledge BioRender.com for providing templates used for illustrations. This study is supported by the CRC1366 (Projects B4 and B1) and CRC834 (Project B5 and B13) by the Deutsche Forschungsgemeinschaft, and the Rolf Schwiete Foundation to S.D. The German Cardiovascular Research network DZHK supports W.T.A., Y.M., M.H.S, S.-I.B., and S.D. S.D. was supported by a fellowship of the Australian Academy of Science. R.P.H. acknowledges research support from the National Health and Medical Research Council of Australia (NHMRC; APP1118576, 1074386), the Australian Research Counsel (ARC; SR110001002), Foundation Leducq Transatlantic Networks of Excellence in Cardiovascular Research (15 CVD 03, 13 CVD 01) and the New South Wales Government Department of Health.

## Author contributions

L.S.T. performed bioinformatic analyses, performed in vitro and in vivo experiments, qPCR, immunohistochemistry, FACS and data analysis. S.F.G. performed in vitro experiments, qPCR and FACS analysis. A.F. performed in vivo experiments. J.W., S.I.B., and I.F. performed and analyzed metabolic analysis. M.M.-R., G.L., and Y.M. performed immunohistochemistry experiments. C.S. and T.A. performed EPIC array. N.B., K.K., M.L., and M.H.S. analyzed EPIC array data. D.J. supported bioinformatic analysis. R.P. and R.P.H performed and analyzed ligand-receptor analysis. E.-M.R., E.F., M.F., and N.R. provided sequencing data and analysis strategies. W.T.A. conducted single-cell library generation. L.S.T., W.T.A., and S.D. designed experiments. L.S.T. and S.D. wrote the manuscript.

## Funding

## Competing interests

The authors declare no competing interests.
