## [Peer Review File · Nature Communications]

REVIEWER COMMENTS

Reviewer #1 (Remarks to the Author):

In their manuscript entitled “Single cell sequencing reveals endothelial plasticity with transient mesenchymal activation after myocardial infarction”, Tombor and co-workers use a combination of endothelial lineage tracing and single cell sequencing to investigate the contribution of endothelial plasticity to post-infarct remodeling. They find that in the post-infarct environment transient Endothelial-Mesenchymal transition is occurring, which may contribute to the regeneration of a vascular network. The experiments are well-designed and innovative and would appeal to a broad audience.

Comments to the authors:

Endothelial cells that associate with a mesenchymal fate show decreased fatty acid metabolism, yet these cells appear highly proliferative and thus in need of an energy source. Did the authors look into the energy metabolism of these cells and found that the “mesenchymal EC” are more glycolytic? Pertinent literature suggest that the shift from FA metabolism to glycolysis may be sufficient to drive EndMT.

Probability values for gene enrichment analysis is unclear from the figures, albeit the text states that gene sets are significantly enriched. From the figure legend, it is unclear if significance is mentioned for individual genes or for the complete gene set. Please modify the figure to include statistical detail or add probability values in the text.

From their sequencing experiment, the authors provide a convincing case for the transient nature of endothelial cells, however it cannot be excluded completely that cells undergoing EndMT would undergo apoptosis and are replaced by proliferating EC. How can the authors exclude this option? Albeit the trajectory State analysis would suggest a transient nature, trajectory state analyses are based on algorithms explaining phenotype data of a single cell type and do not take into account massive replacement. Do the authors have in vivo data on cell death and proliferation (non genetic)? Adding the data of the Confetti mice treated with Galunisertib would surely be instrumental.

The sequencing data suggest that EC that undergo EndMT or EndMA switch to glycolysis, albeit metabolic profiling of these mice (or EC cultures) has not been performed. Is cell metabolism really

altered in these EC or is the gene signature merely illustrative of altered signaling? If a metabolic switch is indeed driving EndMT/EndMA, this would open new therapeutic avenues.

In their in vitro EndMT models, the authors claim that withdrawal of TGFb2 reverts EC back to their normal phenotype, however, TGFb2 is known to suppress proliferation of EC. How can the authors discriminate between increased proliferation of naive EC and reversal of EndMT following the withdrawal of TGFb2 in these experiments? The addition of cell cycle inhibitors in these experiments (excluding the possibility of overgrowth of naive EC) might facilitate here. Moreover, reversibility of EndMT is argued to be critically dependent on the time exposed to TGFb2; did the authors check if EC exposed to TGFb2 for 7-14 days also have the ability to revert to their naive phenotype?

The current manuscript is exemplary on the use of bioinformatics to address cell differentiation kinetics in vivo that fuel new hypotheses, but seemingly little insights from the bioinformatic approaches are validated at the protein level. The manuscript would benefit from quantifying cellular metabolism, expression of differentially expressed proteins and cell cycle to underscore the accuracy of the bioinformatic approaches.

Reviewer #2 (Remarks to the Author):

Utilizing current advancements in mouse genetics and bioinformatics, Tombor et al. present important, novel findings for understanding how the endothelium responds and adapts to a cardiac insult. Furthermore, they provide clarification for misconceptions in the field about “partial endMT” in adult wound healing, and what it actually means in the context of cardiac endothelial cell biology. Several points should be addressed to boost the impact of the manuscript.

1. The paper is essentially written in two tones, which muddles some conclusions: one in support of EndMT (which was the goal, as the intro states), and then a tone for transient activation or “EndMA” (which is data-driven). For this type of manuscript, and in order to more effectively clear up controversy in the literature, the authors should consider phasing out inaccurate use of EndMT (especially in the Results), rather than just the last sentence. The Abstract and especially the Introduction should more directly address the misconceptions to effectively communicate the importance of these findings.

- a. It is worth reinforcing that the data support the cancer literature in establishing the existence of different “activation states” of ECs, beyond the VEGF-dependent tip/stalk EC paradigm.
- b. A big-picture schematic or diagram would help with these clarifications.
2. Throughout the figures, non-EndMA+ cell populations are addressed as “Other” or “EndMA-“ or “homeostatic” – but the term “naïve” is used frequently in the manuscript. Using naïve to describe the cells is confusing; it would be better to use nomenclature from the figures.
3. In the Introduction, the authors state that ECs “actively maintain a quiescent phenotype...” Please provide a reference or new word choice as it may not be correct to state that endothelial cells are quite. They actively maintain a vessel lumen wall, cell polarity, cell and matrix adhesion properties, and respond at all times to changes in flow or chemical signals. “Quiescent” may not be the best terminology.
4. It would be helpful to see basic histology (picosirius red, CD31, the Cdh5-GFP signal) of the time points analyzed to illustrate what the myocardial endothelial network looks like to correlate with your expression data.
5. Especially with the GFP reporter, was any protein validation performed in terms of immunofluorescence? It is appreciated that other cells also express these endMA proteins, but several could be utilized given the reporter system. It would be helpful to know whether there is any regionality of EndMA+ ECs at the time points analyzed.
- a. The “profound” increase in FSP co-localization in Suppl Fig S4 is weak, and it’s surprising that more non-EC FSP staining isn’t observed. Additional time points would be informative and further help with validation.
6. It would be nice to see any images from the Galunisertib experiment in Confetti mice.

Reviewer #3 (Remarks to the Author):

In their manuscript, Tombor et al. describe studies of endothelial cells (EC) in heart tissue following experimentally induced myocardial infarction in a mouse model. For this, publicly available single-cell RNA-sequencing data from a prior study (Forte et al., 2020, Cell Reports 30, 3149–3163) are re-analysed. The authors present evidence that a portion of the endothelial cells undergo transient epithelial-to-mesenchymal transition, which they call “epithelial-to-mesenchymal activation” (EndMA) to highlight the transient nature of this process. This is complemented by new single-cell RNA-sequencing data in a lineage-traced mouse model to ensure that also cells are captured which might have lost Cdh5 expression. In human umbilical cord endothelial cells (HUVEC), the authors

demonstrate that EndMA as induced by TGF-beta is reversible, and that it does not lead to changes in DNA methylation of key target genes of the process.

The manuscript is carefully written and describes the results in full detail. I have a few comments, though.

1. The original analysis of non-cardiomyocyte cells from heart tissue following myocardial infarction presented with 16 clusters of cells, while the authors in this manuscript come up with 19 clusters. The authors should describe how the two analysis strategies differed, and how their clusters relate to the clusters from the original publication.

2. The authors should present representative images for the experiments with Galunisertib, similar to Fig. 2 of Manavski et al., *Circ Res* 2018.

3. The trajectory analysis (Fig. 2) indicates that the proportion of EC in state 4 increases at d1-d7. However, it seems that simultaneously the total number of EC is substantially reduced (in particular at d1 and d3). Thus, it may be that the total number of EC in state 4 remains constant but just the number of cells in other states is reduced. As it is hard to estimate cell counts from Fig. 2A, the authors should give the absolute counts in addition to the proportions. The authors should discuss whether this is due to activation of mesenchymal programs in EC surviving ischemia, or because EC in state 4 are simply more resistant to hypoxia. Also, the relation of the 5 states in the trajectory analysis to the original clusters Ec1-Ec4 (Fig. 1A) would be interesting.

4. The authors analyze in detail the changes in expression of metabolically active enzymes in EC showing activation of mesenchymal markers (EndMA+) compared to EndMA-negative cells. First, they should be more precise to distinguish fatty acid metabolism from fatty acid signaling. To me, it appears that most genes presented in Fig 3C,D are related to fatty acid catabolism and transport, maybe with the exception of DEGS1. Second, the observed metabolic changes may be a direct consequence of hypoxia, shifting glucose catabolism from oxidative phosphorylation to anaerobic catabolism. In this case, glycolysis would need to be induced, while TCA activity may be temporarily reduced. Nevertheless, the statement of the authors that fatty acid catabolism is transiently reduced remains valid. It would be perfect if the authors could show the time course presented in Fig. 3D separately for EndMA+ and EndMA- cells.

5. The presented analysis of Ligand-Receptor interactions presented in Fig. S5 lacks sufficient details for understanding the results. Here, I would suggest to present ligand-receptor pairs, together with the corresponding cell types, instead of just details for the found ligands in EndMA+ cells.

6. The methylation analysis of transient induction of EndMT in HUVEC cells by TGF-beta only presents results for gene body-localized probes. These may not be indicative of regulatory events, which usually effect CpG islands upstream of (or overlapping) the transcription start site. I suggest to include probes +/- 5kb of the TSS.

7. The clusters 0 through 16 in Fig. 4A should be labeled in the legend to this Figure, as well as clusters 0 through 6 in Fig. 4E (similar to Fig. 1A).

8. The sequencing and methylation data from new experiments described in this manuscript need to be made publicly available, in compliance with NatureSpringer policies. This applies to single-cell RNA sequencing data from Cdh5-CreERT2 mT/mG mice, and to the Illumina EPIC-array methylation data from HUVEC. It would be ideal if the authors also made the R code for the analysis available, to ensure full reproducibility of their analysis.

RESPONSE TO THE REVIEWERS COMMENTS

Response to Reviewer #1:

In their manuscript entitled “Single cell sequencing reveals endothelial plasticity with transient mesenchymal activation after myocardial infarction”, Tombor and co-workers use a combination of endothelial lineage tracing and single cell sequencing to investigate the contribution of endothelial plasticity to post-infarct remodeling. They find that in the post-infarct environment transient Endothelial-Mesenchymal transition is occurring, which may contribute to the regeneration of a vascular network. The experiments are well-designed and innovative and would appeal to a broad audience.

Comments to the authors:

Endothelial cells that associate with a mesenchymal fate show decreased fatty acid metabolism, yet these cells appear highly proliferative and thus in need of an energy source. Did the authors look into the energy metabolism of these cells and found that the “mesenchymal EC” are more glycolytic? Pertinent literature suggest that the shift from FA metabolism to glycolysis may be sufficient to drive EndMT.

Answer: Indeed, mesenchymal marker expressing endothelial cells showed higher levels of expression of glycolytic genes (**Figure 3c and new Figure S8a, right panel**). In addition, we determined endothelial cell metabolism *in vitro* by Seahorse measurements. These data demonstrate that TGF- β 2 stimulation and EndMA is associated with a significant increase in glycolysis (**Figure S10a**). Moreover, ^{13}C glucose tracing confirmed an increased metabolism of glucose upon TGF- β 2 stimulation (**Figure S10b**), which is consistent with the recent studies demonstrating that a shift from FA metabolism to glycolysis drives EndMA (<https://www.ncbi.nlm.nih.gov/pmc/articles/PMC5816688/>). Interestingly, this metabolic adaptation was reversible upon withdrawal of TGF- β 2 (**Figure S10b**).

Probability values for gene enrichment analysis is unclear from the figures, albeit the text states that gene sets are significantly enriched. From the figure legend, it is unclear if significance is mentioned for individual genes or for the complete gene set. Please modify the figure to include statistical detail or add probability values in the text.

Answer: We thank the reviewer for this comment. We used the Enrichr software suite to calculate gene set enrichments only for significantly (adjusted p-value < 0.05) regulated genes (<https://pubmed.ncbi.nlm.nih.gov/27141961/>). We used Enrichr's combined score to report GO term enrichment, which was calculated by the p-value of Fisher's exact test and the z-score of the gene set's deviation to its expected rank. We updated the figure legends and method section accordingly.

From their sequencing experiment, the authors provide a convincing case for the transient nature of endothelial cells, however it cannot be excluded completely that cells undergoing EndMT would undergo apoptosis and are replaced by proliferating EC. How can the authors exclude this option? Albeit the trajectory State analysis would suggest a transient nature, trajectory state analyses are based on algorithms explaining phenotype data of a single cell type and do not take into account massive replacement. Do the authors have in vivo data on cell death and proliferation (non genetic)?

Answer: The reviewer is raising an interesting and important point. To prove that mesenchymal cells can revert to an endothelial cell phenotype and are not simply removed by cell death, we performed collagen-tracing studies. In this experimental set up, mesenchymal cells were labelled by injecting tamoxifen in *Col1a2-CreERT2;mT/mG* mice at day 3 to day 7 after infarction (**Reviewer Figure 2a**). Then we performed single cell sequencing and histology at day 28. We found that indeed GFP⁺ cells were detectable in the endothelial cluster (**Reviewer Figure 2b**), demonstrating that Col1a2 traced cells can become transcriptionally similar as compared to endothelial cells between day 7 and day 28 after infarction. These preliminary results were confirmed by histology showing GFP-expressing cells, which co-express the endothelial marker Isolectin B4. (**Reviewer Figure 2d**)

To gain additional insights to the regulation of apoptosis, we determined the expression of cell death genes in EndMA⁺ versus EndMA⁻ cells. Both cell subsets expressed positive and negative regulators of cell death but the expression was not different between EndMA⁺ and EndMA⁻ cells (**Reviewer Figure 1b**). Moreover, we could not find the mesenchymal marker (Sm22) expressing Caspase3-positive endothelial cells at d3 and d7 after infarction (**Reviewer Figure 1a**).

Finally, we performed *in vitro* studies to address, whether EndMA cells might be dying and might be outcompeted by proliferating endothelial cells. Therefore, we treated TGF- β 2 stimulated cells with a cell cycle inhibitor and assessed the effect on the reversibility after TGF- β 2 withdrawal. However, the cell cycle arrest did not affect reversibility of the process (**Reviewer figure 3**) suggesting that mesenchymal marker expressing endothelial cells are not outcompeted by remaining endothelial cells.

Adding the data of the Confetti mice treated with Galunisertib would surely be instrumental.

Answer: We have added representative images and confirmation of the inhibitory effect of Galunisertib as new **Suppl. Figure S7**.

The sequencing data suggest that EC that undergo EndMT or EndMA switch to glycolysis, albeit metabolic profiling of these mice (or EC cultures) has not been performed. Is cell metabolism really altered in these EC or is the gene signature merely illustrative of altered signaling? If a metabolic switch is indeed driving EndMT/EndMA, this would open new therapeutic avenues.

Answer: We thank the reviewer for this comment. We determined the metabolism of endothelial cells upon induction of EndMA *in vitro*. Indeed, glycolysis and glucose metabolism were significantly induced under EndMA promoting conditions (**Suppl. Figure S10**). Unfortunately, we did not succeed in isolating sufficient endothelial cells from mouse hearts for Seahorse measurements. However, we confirmed the down-regulation of the fatty acid binding protein Fabp4 by immunostainings (**Suppl. Figure S9**).

We agree with the reviewer's notion that targeting the metabolic switch could be an interesting therapeutic strategy. So far, most studies addressed the control of endothelial cell metabolism in the context of tumors, but little is known regarding a potential therapeutic interference with the coronary microcirculation. This certainly deserves future studies.

In their in vitro EndMT models, the authors claim that withdrawal of TGF β 2 reverts EC back to their normal phenotype, however, TGF β 2 is known to suppress proliferation of EC. How can the authors discriminate between increased proliferation of naive EC and reversal of EndMT following the withdrawal of TGF β 2 in these experiments? The

addition of cell cycle inhibitors in these experiments (excluding the possibility of overgrowth of naive EC) might facilitate here. Moreover, reversibility of EndMT is argued to be critically dependent on the time exposed to TGF β 2; did the authors check if EC exposed to TGF β 2 for 7-14 days also have the ability to revert to their naive phenotype?

Answer: As suggested by the reviewer, we have addressed the impact of proliferation by adding a cell cycle inhibitor. As shown in **Reviewer Figure 3**, cell cycle inhibition by supplementing 2.5 μ M vitexicarpin had only a minor effect on the reversibility of the mesenchymal transition. The majority of cells still showed a reversible reduction of the mesenchymal markers after removing the TGF- β 2 medium (**Reviewer Figure 3**).

In addition, we further assessed a longer time exposure to TGF- β 2. As shown in **Suppl. Figure S13f-g**, we demonstrate a reversible response after 14 days (11 days TGF- β 2 treatment + 3 days removal).

The current manuscript is exemplary on the use of bioinformatics to address cell differentiation kinetics in vivo that fuel new hypotheses, but seemingly little insights from the bioinformatic approaches are validated at the protein level. The manuscript would benefit from quantifying cellular metabolism, expression of differentially expressed proteins and cell cycle to underscore the accuracy of the bioinformatic approaches.

Answer: To address the reviewer's comments, we have substantiated our findings by performing the following studies:

- We have detected proliferating endothelial cells by histology in the time course after myocardial infarction (**Suppl. Figure S2**)
- We have confirmed the metabolic response of endothelial cells during mesenchymal transition *in vitro* (**Suppl. Figure S10**)
- We have also confirmed the regulation of proteins involved in fatty acid signaling and show that Fabp4 is indeed reduced in the border zone at day 3 after infarction (**Suppl. Figure S9**)
- We have validated the expression of several EndMA markers that had been used by histology (**Suppl. Figures S5 and S6a-b**) and confirmed EndMA induction by FACS (**Suppl. Figure S6c**).

Together, these data confirm the results of the single cell sequencing study.

Response to Reviewer #2:

Utilizing current advancements in mouse genetics and bioinformatics, Tombor et al. present important, novel findings for understanding how the endothelium responds and adapts to a cardiac insult. Furthermore, they provide clarification for misconceptions in the field about “partial endMT” in adult wound healing, and what it actually means in the context of cardiac endothelial cell biology. Several points should be addressed to boost the impact of the manuscript.

1. The paper is essentially written in two tones, which muddles some conclusions: one in support of EndMT (which was the goal, as the intro states), and then a tone for transient activation or “EndMA” (which is data-driven). For this type of manuscript, and in order to more effectively clear up controversy in the literature, the authors should consider phasing out inaccurate use of EndMT (especially in the Results), rather than just the last sentence. The Abstract and especially the Introduction should more directly address the misconceptions to effectively communicate the importance of these findings.

a. It is worth reinforcing that the data support the cancer literature in establishing the existence of different “activation states” of ECs, beyond the VEGF-dependent tip/stalk EC paradigm.

Answer: We thank the reviewer for this suggestion. We have added the concept of EC activation, specifically mesenchymal activation in the revised introduction section of the manuscript. We also cited several recent publications showing endothelial cell plasticity in tumors.

b. A big-picture schematic or diagram would help with these clarifications.

Answer: We thank the reviewer for this suggestion and have added a scheme in **Figure S15** of the revised manuscript.

2. Throughout the figures, non-EndMA+ cell populations are addressed as “Other” or “EndMA-“ or “homeostatic” – but the term “naïve” is used frequently in the manuscript. Using naïve to describe the cells is confusing; it would be better to use nomenclature from the figures.

Answer: We thank the reviewer for this comment and have re-phrased these sentences.

3. In the Introduction, the authors state that ECs “actively maintain a quiescent phenotype...” Please provide a reference or new word choice as it may not be correct to state that endothelial cells are quite. They actively maintain a vessel lumen wall, cell polarity, cell and matrix adhesion properties, and respond at all times to changes in flow or chemical signals. “Quiescent” may not be the best terminology.

Answer: We thank the reviewer for this comment and have re-phrased this sentence.

4. It would be helpful to see basic histology (picrosirius red, CD31, the Cdh5-GFP signal) of the time points analyzed to illustrate what the myocardial endothelial network looks like to correlate with your expression data.

Answer: We have added histological analysis of the vascular network and fibrosis in **Reviewer Figure 4a-b**.

5. Especially with the GFP reporter, was any protein validation performed in terms of immunofluorescence? It is appreciated that other cells also express these endMA proteins, but several could be utilized given the reporter system. It would be helpful to know whether there is any regionality of EndMA+ ECs at the time points analyzed.
a. The “profound” increase in FSP co-localization in Suppl Fig S4 is weak, and it’s surprising that more non-EC FSP staining isn’t observed. Additional time points would be informative and further help with validation.

Answer: To address the reviewer’s comment, we have added additional stainings and time points in the revised version of the manuscript (**Suppl. Figures S5 and S6**). We have analyzed different regions of infarcted hearts and saw less EndMA+ cells in remote zones (**Suppl. Figure 5b**).

We additionally analyzed Fsp1 expression in isolated EC by FACS, confirming the increase in Fsp1+ EC in mice hearts at day 7 after infarction (**Suppl. Figure S6c**).

6. It would be nice to see any images from the Galunisertib experiment in Confetti mice.

Answer: We have added representative images as **Suppl. Figure S7**.

Response to Reviewer #3:

In their manuscript, Tombor et al. describe studies of endothelial cells (EC) in heart tissue following experimentally induced myocardial infarction in a mouse model. For this, publicly available single-cell RNA-sequencing data from a prior study (Forte et al., 2020, Cell Reports 30, 3149–3163) are re-analysed. The authors present evidence that a portion of the endothelial cells undergo transient epithelial-to-mesenchymal transition, which they call "epithelial-to-mesenchymal activation" (EndMA) to highlight the transient nature of this process. This is complemented by new single-cell RNA-sequencing data in a lineage-traced mouse model to ensure that also cells are captured which might have lost *Cdh5* expression. In human umbilical cord endothelial cells (HUVEC), the authors demonstrate that EndMA as induced by TGF-beta is reversible, and that it does not lead to changes in DNA methylation of key target genes of the process.

The manuscript is carefully written and describes the results in full detail. I have a few comments, though.

1. The original analysis of non-cardiomyocyte cells from heart tissue following myocardial infarction presented with 16 clusters of cells, while the authors in this manuscript come up with 19 clusters. The authors should describe how the two analysis strategies differed, and how their clusters relate to the clusters from the original publication.

Answer: In contrast to the analysis done by *Forte et. al*, we used cellranger "count" outputs and merged and normalized them in Seurat 2.3.4.

We adopted filtering strategies to barcodes, keeping cells with > 400 and < 4500 genes, as well as < 18000 UMIs and < 10% mitochondrial gene content.

We used the first 10 principal components for graph-based clustering with a resolution parameter of 0.6, which is expected to give more clusters than $res = 0.5$, as used in *Forte et al*.

2. The authors should present representative images for the experiments with Galunisertib, similar to Fig. 2 of Manavski et al., Circ Res 2018.

Answer: We have included representative images for the Galunisertib experiment in **Suppl. Figure S7**.

3. The trajectory analysis (Fig. 2) indicates that the proportion of EC in state 4 increases at d1-d7. However, it seems that simultaneously the total number of EC is substantially reduced (in particular at d1 and d3). Thus, it may be that the total number of EC in state 4 remains constant but just the number of cells in other states is reduced. As it is hard to estimate cell counts from Fig. 2A, the authors should give the absolute counts in addition to the proportions. The authors should discuss whether this is due to activation of mesenchymal programs in EC surviving ischemia, or because EC in state 4 are simply more resistant to hypoxia. Also, the relation of the 5 states in the trajectory analysis to the original clusters Ec1-Ec4 (Fig. 1A) would be interesting.

Answer: To address the comment of the reviewer, we provide raw data of the number of cells at each time point. Indeed, a much lower number of cells were available in mice hearts during the first days after infarction (**Reviewer Figure 5d**). If we then assess the number of cells in state 4, we did find a slightly reduced number at day 1 and day 3 as compare to baseline or to later time points, however, non-state 4 cells were even further reduced particularly when comparing baseline and day 1 and day 3 (**Reviewer Figure 5d**). Therefore, the percentage of state 4 cells compared to total endothelial cells is profoundly induced to 69% and 81% at day 1 and day 3, respectively (**Reviewer Figure 5d, percentages below graph**). The reduced number of overall endothelial cells at d1 and d3 likely reflects the fact that at these time points a huge increase in inflammatory cells (40-60%) was observed (**Reviewer Figure 5e**), which reduces the relative number of endothelial cells in the single cell sequencing data set. We did not find major differences in cell death related genes comparing state 4 and non-state 4 cells and/or in EndMA⁺ versus EndMA⁻ cells (**Reviewer Figure 1b**) suggesting that there is not a mere survival advantage of EndMA⁺ cells.

We also traced back the trajectory states to the original clusters shown in Figure 1a (**Reviewer Figure 5a-c**). However, states in endothelial cells did not separate between the cell clusters when using a global clustering of all cells in the dataset. Therefore, we separated cells from EC clusters (6,7,17,18) and re-clustered them by their variable genes (**Suppl. Figure 4a**). In this new clustering, we found that state 4 cells predominantly appeared in clusters 4, 5, and 6 which we identified to express mesenchymal markers (**Suppl. Figure 4b**). These clusters, similar to the state 4 in the trajectory, were enriched for d1-d7 timepoints and showed a higher number of EndMA⁺ cells (**Suppl. Figure 4e-f**). Additionally, we found state 3 mapping

predominantly to cluster 8, which we associated with an inflammatory phenotype. We have added the contribution of the states to this clustering to our manuscript (**Suppl. Figure 4h**).

4. The authors analyze in detail the changes in expression of metabolically active enzymes in EC showing activation of mesenchymal markers (EndMA+) compared to EndMA-negative cells. First, they should be more precise to distinguish fatty acid metabolism from fatty acid signaling. To me, it appears that most genes presented in Fig 3C,D are related to fatty acid catabolism and transport, maybe with the exception of DEGS1. Second, the observed metabolic changes may be a direct consequence of hypoxia, shifting glucose catabolism from oxidative phosphorylation to anaerobic catabolism. In this case, glycolysis would need to be induced, while TCA activity may be temporarily reduced. Nevertheless, the statement of the authors that fatty acid catabolism is transiently reduced remains valid. It would be perfect if the authors could show the time course presented in Fig. 3D separately for EndMA+ and EndMA- cells.

Answer: We agree with the reviewer, that the related genes, namely *Cd36*, *Fabp4* and *Fabp5*, are mainly involved in fatty acid transport. We have modified the wording in the text of the manuscript accordingly.

To gain insights whether the induction of glycolysis is induced independently on hypoxia, we assessed endothelial cell metabolism after TGF- β 2-treatment. Indeed, we could show that glycolysis is significantly induced (**Suppl. Figure S10a**). ^{13}C glucose tracing confirmed these data demonstrating that TGF- β 2 reversibly enhances the use of glucose in glycolysis pathways and TCA cycle. (**Suppl. Figure S10b**).

We also found significant enrichment of the gene set “Hallmark Glycolysis” (M5937) when comparing EndMA⁺ vs. EndMA⁻ regulated genes. Gene set enrichment analysis confirmed that EndMA⁺ cells have a higher expression of glycolysis genes (**Suppl. Figure 8a**).

Finally, highly expressed and significantly regulated genes involved in glucose metabolism are shown in the time course after MI (**Reviewer Figure 6**). Unfortunately, there were too few EndMA⁺ and EndMA⁻ cells the early time points. Therefore, this data set was not powered to perform a separate analysis of the two cell populations over time. This is why we pooled the data for the analysis shown in **Figure 3c**.

5. The presented analysis of Ligand-Receptor interactions presented in Fig. S5 lacks sufficient details for understanding the results. Here, I would suggest to present ligand-receptor pairs, together with the corresponding cell types, instead of just details for the found ligands in EndMA+ cells.

Answer: We have added more details regarding the ligand-receptor interactions in the revised Suppl. Figure S11d.

6. The methylation analysis of transient induction of EndMT in HUVEC cells by TGF-beta only presents results for gene body-localized probes. These may not be indicative of regulatory events, which usually effect CpG islands upstream of (or overlapping) the transcription start site. I suggest to include probes +/- 5kb of the TSS.

Answer: We thank the reviewer for this comment. We agree that regulatory elements such as enhancers and CpG-islands may become DNA methylated as part of regulatory events. Thus, we have performed an in-depth analysis of DNA methylation changes at regulatory elements inside and outside of gene bodies using all measured CpG probes on the methylation array. Interestingly, we observed significant changes in methylation patterns in hundreds of such regulatory elements. The stimulation with TGF- β 2 induced the demethylation in 96% of these regions in a fully reversible manner, when the EndMA inducing medium was replaced by normal medium (**Reviewer Figure 7** and **Suppl. Figure S14d**). To gain first insights into the potential implications of these findings, we predicted putative transcription factor binding sites in these demethylated regions. We found an enrichment for binding sites of various *HOX* transcription factors and important endothelial regulatory transcription factors such as *KLF4*, *ERG*, and *GATA2* (**Reviewer Figure 7b**). This could imply that under TGF- β 2 stimulation, regulatory elements become accessible for these transcription factors to induce otherwise silenced genes or gene networks.

Using a recently developed program to predict genes of the EpiRegio database, which holds predicted target genes of the differentially methylated regulatory elements, which are regulated by the respective regions, we have started to identify the potential down-stream genes that might be regulated by *KLF4*, *ERG* or *GATA2*. We have found a few genes, which are predicted to be controlled by the regulatory elements and are indeed differentially expressed in TGF- β 2-stimulated cells in a reversible manner (**Reviewer Figure 7 b-c**).

We believe that these *in silico* analyses provide novel insights but deserve being further explored and validated, which is beyond the scope of the current manuscript.

7. The clusters 0 through 16 in Fig. 4A should be labeled in the legend to this Figure, as well as clusters 0 through 6 in Fig. 4E (similar to Fig. 1A).

Answer: We thank the reviewer for this comment. We updated the figure legend of **Figure 4** with the respective cell types-to-cluster annotations in the revised manuscript.

8. The sequencing and methylation data from new experiments described in this manuscript need to be made publicly available, in compliance with NatureSpringer policies. This applies to single-cell RNA sequencing data from Cdh5-CreERT2 mT/mG mice, and to the Illumina EPIC-array methylation data from HUVEC. It would be ideal if the authors also made the R code for the analysis available, to ensure full reproducibility of their analysis.

Answer: All data will be made available at GEO DataSets upon acceptance of the manuscript. A placeholder is included in the method section. Code will be provided in a GitHub repository. In case the reviewer would like to have access to the data now, we are happy to provide the data via the Nature Communication Office but we hope that the reviewer understands that we would not like to have the data publicly available before acceptance of the manuscript.

a

b Regulation of cell death

Reviewer Figure 1 - EndMA cells and apoptosis marks

a Immunohistochemistry of border zones at d3 or d7 stained for Hoechst (blue), mesenchymal marker Sm22 (green), apoptosis marker Caspase3 (red) and endothelial marker Isolectin B4 (white). Sections were blocked for 1h in block buffer (3% BSA, 5% donkey serum, 0.1% Triton-X100) and incubated overnight at 4°C with primary antibodies for Sm22 (1:100, ab10135, abcam) Isolectin B4 (1:50, B1205, Vector) and cleaved Caspase3 (1:100, 9661, CST). Secondary antibodies were incubated for 1h at room temperature (anti-goat Alexa 488, A11055 Life technologies; donkey anti-rabbit Alexa Flour 555, A31572 Life technologies; SAV Alexa Flour 647, S32357, Invitrogen, all 1:200) and Hoechst 33342 (1:400, AS83218, Ana Spec). Scale bars represent 50 μm. **b** GO-Term enrichment of genes differential regulated between EndMA+ and EndMA- cells (Figure 3a). Genes were classified by their association to regulate cell death: positive regulation (GO:0010942) or negative regulation (GO:0060548).

Reviewer Figure 2 - Tombor et al.

Reviewer Figure 2 - Col1a2-CreERT2 tracing confirms presents of endothelial cells that underwent EndMA after d28

a Schematic illustration of experimental design. We used tamoxifen inducible Col1a2-CreERT2;mT/mG adult (12-weeks, $n = 1$) and old (18 months, $n = 1$) mice to trace cells that express mesenchymal marker *Col1a2* after MI. Tamoxifen was injected i.p. daily from d3-d7 after MI. Hearts were collected at d28 after infarct and processed for single cell RNA sequencing (see methods section for *Cdh5*-CreERT2;mT/mG experiments) or histology. Age-matched littermates ($n = 1$ each) without LAD surgery were used as control. **b** tSNE plot showing GFP positive cells (green). GFP positive cells were predominantly found in mesenchymal (Mes) clusters, but also in endothelial (EC) clusters. **c** tSNE plot showing different endothelial markers (*Vwf*, *Pecam1*, *Cdh5*) and mesenchymal marker *Col1a2* used for determining endothelial and mesenchymal clusters. **d** Border zone of d28 after MI of Col1a2-CreERT;mT/mG hearts. Green signal from anti-GFP (1:400, #GFP-1010, Aves) overlaps with white signal (endothelial marker isolectin B4). Bottom panel represents a zoom of selected area. Scale bars represent 100 μm or 10 μm .

Reviewer Figure 3 - Tombor et al.

Reviewer Figure 3 - Cellcycle inhibition of HUVEC treated with TGF- β 2 does not change reversibility

a Overview of experimental design. HUVEC were seeded in 12-well plates (75.000 cells per well) and cultivated for 24h in control conditions. We treated the cells with TGF- β 2 (see methods sections) and changed the differentiation medium to control medium after 3 days. Cells were then either treated with 2.5 μ M Vitexicarpin or DMSO as control (Reversible). We used cells cultivated for 7 days in control medium (Control) and cells cultivated for 7 days in differentiation medium (EndMA) supplemented with TGF- β 2 as controls. **b** BrdU assay of HUVECs cultivated in control medium and supplemented with 2.5 μ M Vitexicarpin (right panel) or DMSO as control (middle panel) for 24h ($n = 2$). HUVECs were labelled with BrdU for 1h prior cell collection. We used BrdU kit (559619, BD Pharmingen) according to the manufacture's protocol. Cells, which were not treated with BrdU were used as a negative control for defining the gates (left panel). **c** Quantification of BrdU FACS assay. Bars represent the mean relative number of cells found in the G1/G0-phase (blue), G2/M-phase (pink) or S-phase (purple). Vitexicarpin treatment reduced the number of cells in S-phase by 63.89% \pm 4.79% compared to DMSO control. **d-e** qPCR showing reduction of *TAGLN* (SM22) (**d**) and *CNN1* (**e**) levels after withdrawal of differentiation medium. Data was normalized to control and housekeeping genes *GAPDH* and *RPLP0* and shown as fold-change. Error bars represent standard error of the mean of replicates of $n = 3$ experiments. Cells in reversible + DMSO had a significant decrease in *TAGLN* (97.10% \pm 0.18%) and *CNN1* (99.5% \pm 0.05%) expression compared to EndMA. Samples with cell cycle inhibitor Vitexicarpin had a similar decrease in *TAGLN* (81.91% \pm 0.61%) and *CNN1* (91.89% \pm 0.84%).

Reviewer Figure 4 - Basic histology of heart sections after infarct

a Picosirius red staining of sections after myocardial infarct. Collagen type I (red) increases over time. **b** Immunohistochemistry staining of heart sections after myocardial infarct. We used anti-GFP (green) (1:400, #GFP-1010, Aves) and anti-Pecam1 (white) (1:50, 2383-20B-500, VWR) as primary antibodies and anti-chicken IgY (#F-1005, Aves) and anti-rat IgG (A-21247, Thermo fisher, both 1:200) as secondary antibodies.

Reviewer Figure 5- Tombor et al.

Reviewer Figure 5 - Endothelial states in endothelial clustering

a tSNE plot showing states identified by trajectory (Figure 2a) mapped back to the original clustering shown in Figure 1a. Cells in grey were not used for trajectory analysis as they were not double positive for *Pecam1* and *Cdh5*. **b** tSNE plot of original EC clusters (6,7,17,18) as identified in Figure 1b. **c** Bar plot showing the number of cells associated with original clusters for all trajectory states. **d** Bar plot showing the absolute number of cells contributing to state 4 (blue) compared to non-state 4 cells (grey bars). For d1-d7 the ratio of cells in state 4 compared to other cells was significantly enriched ($p < 0.05$, Chi-Squared test with Yates correction, compared to baseline). The relative number of cells are plotted below. **e** Pie charts showing changes in the relative contribution of the three major cell types in the dataset (endothelial cells, fibroblasts (Fb), monocytes (Mono)). While the number of cells per timepoint in total stays within the range of $n = 6568$ to $n = 3453$ at d1, the relative contribution of monocytes increases from 5% (homeostasis) to 63.9% (d3) and decreases again to 5.5% (d28). Inversely, relative numbers of ECs and Fbs decrease and increase at later stage.

Reviewer Figure 6 - Tombor et al.

Reviewer Figure 6 - Glycolysis associated genes in EndMA+ and EndMA- (EC) subsets over time

Bar plots of glycolysis associated genes presented in Figure 3c. Data shows mean UMI values for *Gapdh*, *Hk1*, *Pkm* and *Aldoa* of single cell RNA-seq experiment (Forte et al. 2020). Cells were classified as EndMA+ (orange) as described in Figure 3. We pooled early timepoints (d1-d5), d7 and late timepoints (d14-d28). Error bars represent standard error of the mean (SEM).

Reviewer Figure 7 - Tombor et al.

chromosome region	REM ID	associated gene	ENSEMBL ID	Control to DM	Change (mean difference log2)	DM to DM+FM	Change (mean difference log2)	Predicted TF binding
chr2:150586195-150586604	REM1342527	RND3	ENSG00000115963	demethylated	1.42	methylated	-1.48	GATA2
chr6:36724750-36724849	REM1949737	CDKN1A	ENSG00000124762	demethylated	0.75	methylated	-0.91	ERG
chr7:132072152-132072901	REM2124648	PLXNA4	ENSG00000221866	demethylated	0.43	methylated	-0.34	KLF4
chr12:49276796-49278405	REM0483399	TUBA1A	ENSG00000167552	demethylated	0.59	methylated	-0.64	KLF4

Reviewer Figure 7 - DNA methylation in enhancer elements

a Heatmap showing mean DNA methylation of regulatory elements (REMs) in Control, TGF- β 2 (3d) and -TGF- β 2 (7d) (see figure **S14**). 848,254 CpG probes that were retained after EPIC array quality filtering were overlapped with 2.4 million annotated REMs of the *EpiRegio* database¹ and analysed using *RnBeads* software. CpG methylation was averaged over all CpG probes overlapping a REM. Differential methylated regions with a mean methylation difference ≥ 0.1 between Control and + TGF- β 2 (3d) or + TGF- β 2 (3d) and - TGF- β 2 (7d) respectively, and a FDR corrected p -value ≤ 0.05 were selected. We found 454 REMs being differential methylated in both comparisons (Control vs. +TGF- β 2 (3d) and +TGF- β 2 (3d) vs. -TGF- β 2 (7d)). These REMs were clustered by their DNA methylation pattern using hierarchical clustering. Among these differentially regulated REMs, the majority (96%, upper part of the heatmap) followed a pattern of less methylation in + TGF- β 2 (3d) but increased methylation levels to baseline conditions after withdrawal to control medium. **b** Transcription factors ranked by their p -value (FDR < 0.01) after motif enrichment analysis in REMs following the pattern of less methylation in + TGF- β 2 (3d) and regain of methylation in - TGF- β 2 (7d) (see **a**). To perform the motif enrichment analysis, we applied PASTAA² using 633 human TF motifs from the JASPAR database (version 2020)³. Among predicted transcription factors we found *GATA2*, *ERG* and *KLF4*, which are known regulators of endothelial identity or function. **c** Table showing examples of REMs with putative binding motifs for *GATA2*, *ERG* and *KLF4* annotated using FIMO⁴. Shown genes were demethylated upon + TGF- β 2 (3d) and regained methylation levels upon - TGF- β 2 (7d). **d** FPKM values taken from *Glaser et al. 2020* (GSE143148) bulk-RNA sequencing of Control and + TGF- β 2 (3d) samples for genes shown in (**c**). All genes showed significantly (adjusted $p < 0.05$) higher expression among TGF- β 2 treatment compared to control

¹ Baumgarten N, Hecker D, Karunanithi S, Schmidt F, List M, Schulz MH. EpiRegio: analysis and retrieval of regulatory elements linked to genes. *Nucleic Acids Res.* 2020;48(W1):W193-W199. doi:10.1093/nar/gkaa382

² Roeder HG, Manke T, O'Keeffe S, Vingron M, Haas SA. PASTAA: identifying transcription factors associated with sets of co-regulated genes. *Bioinformatics.* 2009;25(4):435-442. doi:10.1093/bioinformatics/btn627

³ Fornes O, Castro-Mondragon JA, Khan A, et al. JASPAR 2020: update of the open-access database of transcription factor binding profiles. *Nucleic Acids Res.* 2020;48(D1):D87-D92. doi:10.1093/nar/gkz1001

⁴ Grant CE, Bailey TL, Noble WS. FIMO: scanning for occurrences of a given motif. *Bioinformatics.* 2011;27(7):1017-1018. doi:10.1093/bioinformatics/btr064

REVIEWERS' COMMENTS

Reviewer #1 (Remarks to the Author):

The authors have provided additional data that deals with the reviewers comments sufficiently and have improved the quality of their manuscript.

However, albeit the authors found the reviewers comments valuable enough to perform confirmatory experiments, the authors failed to detail these findings in their manuscript. I strongly suggest to include the findings discussed in reviewer figure 1-3 in the manuscript and include these figures as data supplement.

Confirming that the endothelial-derived mesenchymal cells can revert back to an endothelial cell fate - and not go into apoptosis, have a proliferative disadvantage or reversal is time-dependent - is essential to understand the value of the data presented.

Reviewer #2 (Remarks to the Author):

Revisions were adequate

Reviewer #3 (Remarks to the Author):

The authors have invested a lot of work to answer my comments and have addressed almost all of my points. There are only two things left:

1. It would be great if readers could assess the correspondence between the clusterings by Forte and al. and the one presented in the actual manuscript (w.r.t. my point #1). This could be presented, e.g., by a Sankey diagram provided in the Supplemental Material.

2. I appreciate the efforts that the authors invested into the Figures for Reviewers. It would be great if these figures could be included in the final material, either as further Supplemental Material or, if the reviews and rebuttals are being published, they should be contained there.

Response to reviewer

Reviewer #1 (Remarks to the Author):

The authors have provided additional data that deals with the reviewers comments sufficiently and have improved the quality of their manuscript.

However, albeit the authors found the reviewers comments valuable enough to perform confirmatory experiments, the authors failed to detail these findings in their manuscript. I strongly suggest to include the findings discussed in reviewer figure 1-3 in the manuscript and include these figures as data supplement.

Confirming that the endothelial-derived mesenchymal cells can revert back to an endothelial cell fate - and not go into apoptosis, have a proliferative disadvantage or reversal is time-dependent - is essential to understand the value of the data presented.

We thank the reviewer for the comments. We have included the reviewer figures 1-3 into the supplementary data (supplementary figure 7, 8 and 16).

Reviewer #3 (Remarks to the Author):

The authors have invested a lot of work to answer my comments and have addressed almost all of my points. There are only two things left:

1. It would be great if readers could assess the correspondence between the clusterings by Forte and al. and the one presented in the actual manuscript (w.r.t. my point #1). This could be presented, e.g., by a Sankey diagram provided in the Supplemental Material.

We thank the reviewer for pointing us to this issue. We have included a Sankey diagram into the Supplementary Material (supplementary figure 1b). The comparison largely showed a match between cell types annotated in the manuscript and in Forte et al.

2. I appreciate the efforts that the authors invested into the Figures for Reviewers. It would be great if these figures could be included in the final material, either as further Supplemental Material or, if the reviews and rebuttals are being published, they should be contained there.

We thank the reviewer for addressing the value of the data created during the revision process. We have included the reviewer figures 1-3 into the supplementary material and are willing to publish rebuttals and reviews to show the additional figures there.